# Transcriptionally active enhancers in human cancer cells

Katja Lidschreiber[1,2,†] (ID), Lisa A Jung[2,3] (ID), Henrik von der Emde[1] (ID), Kashyap Dave[4] (ID), Jussi Taipale[4,5,6] (ID), Patrick Cramer[1,2,*] (ID) & Michael Lidschreiber[1,2,**,†] (ID)

## Abstract

**The growth of human cancer cells is driven by aberrant enhancer and gene transcription activity. Here, we use transient transcriptome sequencing (TT-seq) to map thousands of transcriptionally active putative enhancers in fourteen human cancer cell lines covering seven types of cancer. These enhancers were associated with cell type-specific gene expression, enriched for genetic variants that predispose to cancer, and included functionally verified enhancers. Enhancer–promoter (E–P) pairing by correlation of transcription activity revealed ~ 40,000 putative E–P pairs, which were depleted for housekeeping genes and enriched for transcription factors, cancer-associated genes, and 3D conformational proximity. The cell type specificity and transcription activity of target genes increased with the number of paired putative enhancers. Our results represent a rich resource for future studies of gene regulation by enhancers and their role in driving cancerous cell growth.**

**Keywords** cancer; enhancers; eRNAs; transcription; TT-seq
**Subject Categories** Cancer; Chromatin, Transcription & Genomics
**Mol Syst Biol. (2021) 17: e9873**

## Introduction

In metazoans, the spatiotemporal regulation of gene expression governs development and homeostasis, and its misregulation contributes to a wide range of human diseases. The underlying cell type-specific gene expression programs are controlled by enhancer elements, which are short segments of *cis*-regulatory DNA (Bulger & Groudine, 2010). Enhancers are typically 100 base pairs (bp) to 1 kbp in length, contain binding sites for sequence-specific transcription factors, and enable regulation of target gene transcription (Furlong & Levine, 2018). The activity of enhancers is to a large extent cell type specific, and tens of thousands of enhancers are thought to be active in any one cell type (Heintzman *et al*, 2009; Dunham *et al*, 2012; Andersson *et al*, 2014).

Gene dysregulation is a hallmark of cancer and is caused by aberrant combinations of transcription factors, co-factors, and enhancers, which change transcriptional networks and cause a pathological cell state (Sur & Taipale, 2016; Bradner *et al*, 2017; Wouters *et al*, 2017). Recently, several studies of transcription dysregulation in tumor cells have revealed how dysfunctions of enhancers contribute to tumor risk, tumorigenesis, progression, and survival (Cohen *et al*, 2017; Chen *et al*, 2018; Chu *et al*, 2018; Corces *et al*, 2018). Enhancer dysfunction can be caused in *cis* by mutations in enhancers that alter their activity or target gene specificity. However, more commonly, enhancer dysfunction is caused in *trans* by mutations in transcription factors, chromatin remodelers, or DNA modifiers that can also alter enhancer activity (Sur & Taipale, 2016). Both, *cis* and *trans* mutations, can lead to loss of tumor-suppressive enhancers and activation of oncogenic enhancers. A current question in cancer genomics is therefore to understand how somatic mutations in *cis* and in *trans* alter transcription programs to cause a cancer phenotype (Kolch *et al*, 2015), and addressing it requires quantitative analysis of the activity of enhancers in human cancer cells.

Enhancers act in the recruitment of RNA polymerase II (Pol II) to target gene promoters and stimulate gene transcription (Levine, 2010; Spitz & Furlong, 2012; Vernimmen & Bickmore, 2015; Li *et al*, 2016; Field & Adelman, 2020). Enhancer function depends on their DNA sequence, local chromatin modifications, binding of transcription factors, and transcriptional co-activators. Enhancers can be located upstream or downstream of their target genes, or within introns. Moreover, they can be located at large distances from their target gene, even 1 Mbp away. The putative location of enhancers may be identified through genome-wide mapping of histone modifications that are enriched in enhancer regions, in particular acetylation of histone H3 at residue lysine 27 (H3K27ac), and monomethylation

1 Department of Molecular Biology, Max Planck Institute for Biophysical Chemistry, Göttingen, Germany
2 Department of Biosciences and Nutrition, Karolinska Institutet, NEO, Huddinge, Sweden
3 Department of Cell and Molecular Biology, Karolinska Institutet, Biomedicum, Solna, Sweden
4 Department of Medical Biochemistry and Biophysics, Karolinska Institutet, Biomedicum, Solna, Sweden
5 Department of Biochemistry, University of Cambridge, Cambridge, UK
6 Genome-Scale Biology Program, University of Helsinki, Helsinki, Finland
  *Corresponding author. Tel: +49 551 201 2800; E-mail: pcramer@mpibpc.mpg.de
  **Corresponding author. Tel: +49 551 201 2806; E-mail: michael.lidschreiber@mpibpc.mpg.de
  † These authors contributed equally to this work

of H3 lysine 4 (H3K4me1), which often coincides with the absence of the trimethylated form of H3 lysine 4 (H3K4me3) (Heintzman et al, 2007; Heintzman et al, 2009; Rada-Iglesias et al, 2011). Active enhancers are also typically located within open chromatin regions, which can be identified by DNase-seq (DNase I digestion and sequencing) or ATAC-seq (assay for transposase-accessible chromatin with high-throughput sequencing) (Boyle et al, 2008; Buenrostro et al, 2013).

Enhancers can themselves be transcribed by Pol II, which produces enhancer RNA (eRNA) (Tuan et al, 1992; Ashe et al, 1997; Routledge & Proudfoot, 2002; de Santa et al, 2010; Kim et al, 2010; Koch et al, 2011). eRNAs are non-coding RNAs with an average length of 1,000–5,000 nucleotides (nt) (de Santa et al, 2010; Kim et al, 2010; Schwalb et al, 2016). They are generally unspliced, not polyadenylated, and short-lived due to their rapid degradation by the nuclear exosome (de Santa et al, 2010; Kaikkonen et al, 2013; Andersson et al, 2014; Core et al, 2014; Lubas et al, 2015; Schwalb et al, 2016). The possible functions of enhancer transcription and eRNAs remain unclear (Natoli & Andrau, 2012; Li et al, 2016; Lewis et al, 2019). Enhancer transcription might be a by-product of random engagement of Pol II with open chromatin. Alternatively, the process of enhancer transcription may help in creating or maintaining an open and active chromatin environment by recruiting chromatin remodelers through their association with transcribing Pol II (Wilson et al, 1996; Cho et al, 1998; Gribnau et al, 2000; Kaikkonen et al, 2013). It is also possible that eRNAs have functional roles in target gene transcription because depletion of specific eRNAs has been reported to decrease mRNA transcription (Lam et al, 2013; Melo et al, 2013; Schaukowitch et al, 2014; Carullo et al, 2020). eRNAs may also facilitate enhancer–promoter looping (Lai et al, 2013; Li et al, 2013; Hsieh et al, 2014) or the recruitment of transcription factors (Sigova et al, 2015).

eRNAs can be used to map active enhancers (de Santa et al, 2010; Kim et al, 2010; Wang et al, 2011; Kaikkonen et al, 2013; Wu et al, 2014; Arner et al, 2015; Schwalb et al, 2016; Michel et al, 2017). eRNAs as enhancer marks have the advantage to identify enhancers that are transcriptionally active, and this has been related to the transactivating function of enhancers on their target promoters (Kim et al, 2010; Wang et al, 2011; Franco et al, 2015; Hah et al, 2015; Michel et al, 2017). This functional indication from the presence of eRNAs is not evident from other enhancer marks (e.g., open chromatin, histone modifications) (Hah et al, 2013; Zhu et al, 2013; Andersson et al, 2014; Core et al, 2014; Franco et al, 2015; Franco et al, 2018; Henriques et al, 2018; Fitz et al, 2020). However, eRNAs are difficult to detect, due to their rapid degradation and low expression levels. Transcriptomic techniques measuring RNA abundance such as RNA-seq or cap analysis of gene expression (CAGE) can also detect eRNAs (Kodzius et al, 2006; Andersson et al, 2014), but are often less sensitive. Detection of unstable eRNAs is, however, facilitated by techniques that monitor newly synthesized RNA (Core et al, 2014; Mahat et al, 2016; Wouters et al, 2017; Franco et al, 2018; Hirabayashi et al, 2019), such as transient transcriptome sequencing (TT-seq) (Schwalb et al, 2016; Michel et al, 2017). TT-seq combines a short 4-thiouridine (4sU) RNA labeling pulse with a RNA fragmentation step to enrich for newly synthesized RNA, and thereby detects short-lived transcripts with high sensitivity. TT-seq monitors transcription genome wide in an unbiased manner and thus measures transcription at both enhancers and their target genes

simultaneously (Schwalb et al, 2016; Michel et al, 2017; Gressel et al, 2019b; Żylicz et al, 2019). TT-seq is therefore ideally suited to map transcriptionally active enhancers in cancer cells and to study gene regulation in cancer.

Here, we use TT-seq to map thousands of transcriptionally active, putative enhancers in fourteen human cancer cell lines covering seven types of cancer. For the majority of cell lines, this represents the first comprehensive genome-wide map of enhancer transcription. We conducted an in-depth analysis of the data, derived thousands of putative enhancer–promoter (E–P) pairs, and extracted general features of enhancers and transcription landscapes in cancer cells. Finally, we provide a comprehensive catalog of transcribed candidate enhancers with cancer-associated somatic mutations and putative enhancer–promoter pairs involving cancer-associated genes. Overall our results serve as a resource to study enhancer activity and gene regulation, and to select candidate enhancers for functional studies.

## Results

### Mapping transcription in human cancer cells

We applied transient transcriptome sequencing (TT-seq (Schwalb et al, 2016)) to map transcription in fourteen human cancer cell lines representing seven different cancer types: brain (M059J, U-118 MG), breast (BT-20, MDA-MB-231), colorectal (GP5d, HCT 116, LoVo), kidney (769-P, 786-O), prostate (DU 145, PC-3), and uterine cancer (SK-UT-1) as well as leukemia (Jurkat (Michel et al, 2017), K562 (Schwalb et al, 2016)). TT-seq samples comprising newly synthesized RNA and total RNA, i.e., RNA-seq, were collected as independent biological duplicates, and newly synthesized RNA labeling and isolation was carried out as described (Schwalb et al, 2016) with minor modifications (Fig 1A; see Materials and Methods for details). TT-seq libraries were sequenced to a depth of ~ 150 million uniquely mapped read pairs (110–232 million per sample). The obtained data sets were highly reproducible (Spearman's rho > 0.98; Appendix Fig S1A).

To identify genomic intervals of uninterrupted transcription (transcription units, TUs) in our data sets, we used the hidden Markov model-based genome segmentation algorithm GenoSTAN (Zacher et al, 2017) (Materials and Methods). TUs overlapping annotated protein-coding genes were defined as mRNAs. All other TUs were considered non-coding (nc) RNAs and were further classified according to their genomic location relative to protein-coding genes into four categories: upstream antisense RNA (uaRNA), convergent RNA (conRNA), antisense RNA (asRNA), and intergenic RNA (Fig 1B, Materials and Methods). On average, GenoSTAN identified ~ 44,000 TUs per cell line (ranging from 26,729 to 64,843), where of ~ 11,000 were classified as mRNAs (Fig EV1A), and the remaining TUs as ncRNAs (uaRNA, conRNA, asRNA, and intergenic RNA).

### Enhancer transcription at thousands of genomic sites

To identify ncRNAs originating from enhancer regions, we first restricted the set of ncRNAs to those originating outside of promoter regions (TSS ± 1 kb, i.e., asRNAs and intergenic RNAs). We then made use of a large set of putative enhancer regions (Zacher et al,

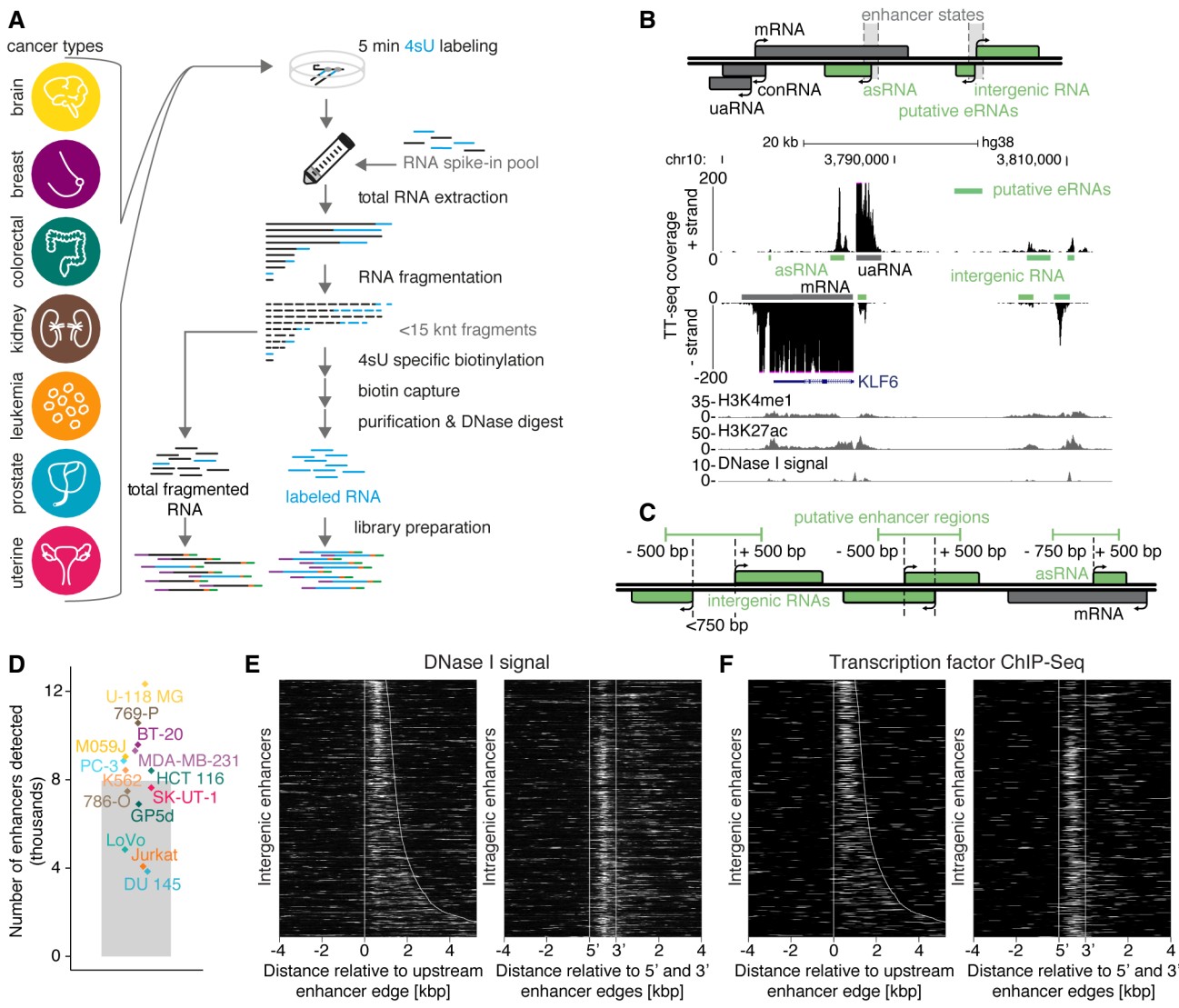

**Figure 1. TT-seq analysis and transcript annotation in human cancer cells.**

A Experimental design. TT-seq data of fourteen human cell lines representing seven different cancer types: brain (M059J, U-118 MG), breast (BT-20, MDA-MB-231), colorectal (GP5d, HCT 116, LoVo), kidney (769-P, 786-O), prostate (DU 145, PC-3), and uterine cancer (SK-UT-1) as well as leukemia (Jurkat (Michel *et al*, 2017), K562 (Schwalb *et al*, 2016). For TT-seq, newly synthesized RNA is metabolically labeled with 4-thiouridine (4sU) for five minutes and labeled RNA is isolated, and processed as described (Schwalb *et al*, 2016) with minor modifications ((Gressel *et al*, 2019a) Materials and Methods). Libraries of total fragmented RNA (RNA-seq) and labeled RNA are sequenced.

B Definition of transcript classes and putative enhancer RNA (eRNA) annotation. Top: non-coding RNAs are classified into four categories: upstream antisense RNA (uaRNA), convergent RNA (conRNA), antisense RNA (asRNA), and intergenic RNA. asRNAs and bidirectional intergenic RNAs originating from GenoSTAN enhancer state regions (Zacher *et al*, 2017) are classified as putative eRNAs (green). Bottom: exemplary UCSC genome browser view of TT-seq coverage in HCT 116 colorectal cancer cells and transcript annotation at the *KLF6* locus (hg38; chr10:3,758,830–3,815,253 (Kent *et al*, 2002)). Due to the high expression of *KLF6*, the TT-seq coverage is cut at 200 to allow for better visualization of the surrounding eRNA signal. H3K4me1 ChIP-seq, H3K27ac ChIP-seq, and DNase I-seq signal from ENCODE (Dunham *et al*, 2012).

C Definition of putative enhancer regions. For non-overlapping (left) and overlapping (middle) bidirectional intergenic RNAs, the region between the TSSs of each pair is extended by 500 bp in both directions, and defined as putative intergenic enhancer region. For asRNAs (right), the region covering 750 bp upstream of the TSS to 500 bp downstream is defined as putative intragenic enhancer region.

D Number of putative enhancer regions per cell line. Bar depicts the mean. Diamonds represent individual numbers in each cell line and are color-coded according to cancer type (as in (A)).

E Heatmaps of DNase I signal (Dunham *et al*, 2012) for enhancer regions (rows). For intergenic enhancers (left), the signal is aligned at the upstream (relative to + strand) enhancer edge, shown from 4 kbp upstream to 5 kbp downstream, and ordered by increasing length of enhancer regions. For intragenic enhancers (right), the signal was aligned at the upstream and downstream enhancer edges and shown within ± 4 kbp. Exemplary data from LoVo colorectal cancer cells.

F Heatmaps as in (E) showing composite ChIP-seq peak coverage from 326 TFs (Yan *et al*, 2013).

2017) that were predicted from patterns of five histone modifications (H3K4me1, H3K4me3, H3K36me3, H3K27me3, and H3K9me3) in 127 cell types and tissues (Dunham et al, 2012; Kundaje et al, 2015). asRNAs and intergenic RNAs originating from this set of putative enhancer regions were classified as putative eRNAs (Fig 1B, Materials and Methods). Moreover, we restricted the set of intergenic eRNAs to only those originating from regions with TT-seq detected transcription on both strands (i.e., bidirectional intergenic RNAs). This annotation strategy led to the identification of on average ~ 9,000 (4,357–14,695) putative eRNAs per cell line (Fig EV1A).

Compared to the set of ncRNAs not overlapping enhancer state regions, our putative eRNAs (hereafter also referred to as "eRNAs") clearly showed properties characteristic of actively transcribed enhancers (Fig EV1B–D, Appendix Fig S1B–E). The region surrounding the TSS of eRNAs was enriched for the active enhancer mark H3K27ac (Fig EV1B) and for transcription factor binding as monitored by ChIP-seq (Fig EV1C). Moreover, eRNAs were characterized by shorter RNA half-lives (Fig EV1D), which we calculated as described (Schwalb et al, 2016).

Based on our eRNA annotation, we next defined putative enhancer regions, which are regions from which eRNA synthesis is initiated. To define intergenic enhancer regions, we selected the region between the TSSs of each pair of intergenic bidirectional eRNAs and extended it by 500 bp in both directions (Fig 1C, Materials and Methods). To define intragenic enhancer regions, we selected the region covering 750 bp upstream to 500 bp downstream of the TSS of antisense eRNAs. This annotation strategy led to the identification of on average ~ 8,000 (3,848–12,337) putative enhancer regions per cell line (Fig 1D), of which about half were located within protein-coding genes (Appendix Fig S1B). The resulting putative enhancer regions (hereafter also referred to as "enhancer regions" or "enhancers") were highly concordant with DNase I-hypersensitive sites (Fig 1E) and transcription factor binding regions identified by ChIP-seq (Fig 1F), and overlapped with enhancer annotations derived from CAGE or ChRO-seq data (Appendix Fig S1F; (Andersson et al, 2014; Chu et al, 2018)). As expected, TT-seq was more sensitive in calling transcribed enhancers than CAGE. These efforts identified thousands of putative eRNAs and enhancers in fourteen human cancer cell lines (Table EV1).

## Transcription directionality is preserved across cell types

The majority of eukaryotic promoter regions are divergently transcribed with the TSS for the protein-coding gene being accompanied by an upstream, antisense TSS producing often non-coding, upstream antisense RNAs (uaRNAs) (Core et al, 2008; Seila et al, 2008; Neil et al, 2009; Almada et al, 2013). Many active enhancers are also divergently transcribed (de Santa et al, 2010; Kim et al, 2010) producing eRNAs from both strands. Since uaRNAs and eRNAs are rapidly degraded (Flynn et al, 2011; Lubas et al, 2015; Schwalb et al, 2016), a measure of RNA synthesis, as provided by TT-seq, is needed to precisely quantify transcription directionality. We therefore investigated bidirectional transcription at promoters and putative enhancers. For promoters, we counted TT-seq reads in 1 kbp regions downstream (sense) and upstream (antisense) of the protein-coding TSS, and excluded divergent protein-coding gene pairs. For enhancers, we took all bidirectional intergenic enhancer regions shorter than 3 kbp and counted TT-seq reads in this region mapped to the plus and minus strand. Based on this, we defined a promoter directionality score as the ratio of sense and antisense read counts, and an enhancer directionality score as the ratio of plus and minus strand read counts ((Jin et al, 2017), Materials and Methods).

We observed upstream antisense transcription at 68% of active promoters across all cell lines (64–75% per cell line). At these promoters, transcription was generally biased toward the coding direction (median ~ 3.1-fold; Fig 2A), in agreement with previous studies (Core et al, 2008; Seila et al, 2008; Xu et al, 2009; Ntini et al, 2013; Sigova et al, 2013; Core et al, 2014; Mayer et al, 2015; Jin et al, 2017). The overall lower TT-seq signals for uaRNAs compared with mRNAs suggested that the remaining promoters might also be bidirectional but with upstream antisense transcription not being detected at our sequencing depth. Indeed, these seemingly unidirectional promoters were on average about 2.5-fold less transcribed than bidirectional promoters (Fig 2B). To test this further, we asked if seemingly unidirectional promoters in one cell line were detected as bidirectional in one of the other cell lines, in which these particular promoters are more highly transcribed. When considering all cell lines, 82% of promoters showed upstream antisense transcription in at least one of the cell lines. The maximum transcription levels of the remaining 18% seemingly unidirectional promoters were about 4-fold lower than those of bidirectional promoters (Appendix Fig S2A). Next, we correlated mRNA and uaRNA synthesis within each cell line and did observe only a weak correlation ($r = 0.25$; Appendix Fig S2B) in agreement with others (Churchman & Weissman, 2011) and with the observed substantial variability of directionality between promoters (Fig 2A). However, when we correlated uaRNA and mRNA synthesis at bidirectional promoters across cell lines, we observed positive correlations for the majority of promoters (median $r = 0.53$, Fig 2C). 63% of promoters had a correlation coefficient above 0.4, indicating that mRNA and uaRNA synthesis are largely coregulated.

In contrast to promoters, where uaRNAs generally showed less TT-seq signal than their corresponding mRNAs, bidirectional enhancers had no directional preference (Fig 2D). Yet, directionality scores varied strongly and 39% of bidirectional enhancers preferentially transcribed one of the strands (|directionality score| ≥ 3; Fig 2 D). The remaining 61% of enhancers were transcribed more bidirectionally balanced on both strands (|directionality score| < 3; Fig 2 D). When we correlated plus and minus strand transcription at enhancers across cell lines, we observed strong positive correlations for the majority of enhancers (median $r = 0.59$; Fig 2E). 68% of enhancers had a correlation coefficient above 0.4. This indicates that enhancer transcription generally maintains its strand preference between different cell lines.

These observations support divergent transcription initiation as a general feature of promoters and enhancers (Core et al, 2014; Andersson et al, 2015; Andersson & Sandelin, 2020) and indicate that the directionality of divergent transcription at promoters and enhancers is widely preserved across cell types.

## Transcription at functionally verified enhancers

We next asked whether enhancers identified based on their TT-seq-measured eRNA synthesis correspond to enhancers that were

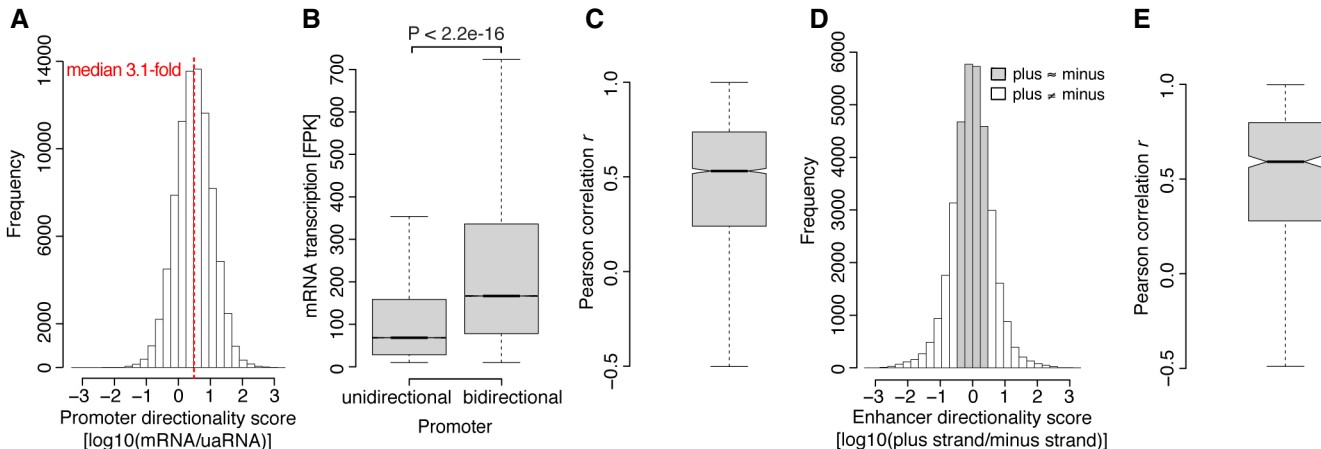

**Figure 2. Transcription directionality is preserved across cell types.**

A   Histogram shows genome-wide distribution of $\log_{10}$-transformed directionality scores at bidirectional promoters. Directionality score for promoters is defined as the ratio of mRNA and uaRNA reads counted within 1 kbp regions downstream sense and upstream antisense of the protein-coding TSS. Promoters with divergently transcribed protein-coding gene pairs were excluded (Materials and Methods). The red dashed line marks the median.

B   Bidirectional promoters show higher mRNA synthesis. Boxplots showing TT-seq signal for mRNAs from unidirectional ($n = 39,376$) and bidirectional ($n = 83,363$) promoters. Bidirectional and unidirectional promoters were defined for each cell line separately (Materials and Methods) and then plotted together. Box limits are the first and third quartiles, the band inside the box is the median. The ends of the whiskers extend the box by 1.5 times the interquartile range. Notches represent 95% confidence intervals for the median values. Outliers not shown. *P*-value by two-sided Mann–Whitney *U*-test.

C   Boxplot showing pairwise Pearson correlation between uaRNA and mRNA synthesis at bidirectional promoters across cell lines. Only promoters with observed bidirectional transcription in at least five cell lines and varying transcription across cell lines (coefficient of variation > 50%) were considered ($n = 3,468$, Materials and Methods). Median, hinges, whiskers, and notches are shown as in (B).

D   Histogram shows genome-wide distribution of $\log_{10}$-transformed directionality scores at bidirectional enhancers. Directionality score for enhancers is defined as the ratio of reads mapped to the plus and minus strand. Only bidirectional intergenic enhancer regions shorter than 3 kbp were considered (Materials and Methods). In accordance with Jin *et al* (2017), enhancer regions were categorized as having a preferred direction (white) if the absolute value of the plus-to-minus strand ratio was ≥ 3 and as bidirectionally balanced (gray) if it was < 3. Note that the distribution would be less balanced if unidirectionally transcribed enhancers were included.

E   Boxplot showing pairwise Pearson correlation between plus and minus strand transcription at bidirectional enhancers over cell lines. Only enhancers with observed bidirectional transcription in at least 5 cell lines and varying transcription over cell lines (coefficient of variation > 50%) were considered ($n = 715$, Materials and Methods). Median, hinges, whiskers, and notches are shown as in (B).

previously observed to be functional in target gene activation. Mansour and colleagues dissected the molecular mechanisms by which a short 12-bp insertion approximately 8 kbp upstream of the *TAL1* oncogene introduces novel binding sites for the transcription factor MYB, and establish a new super-enhancer in a subset of acute lymphoblastic leukemia (ALL) cases (Mansour *et al*, 2014). In our data from the same ALL cell line (Jurkat), *TAL1* is highly expressed and the insertion region is flanked by eRNAs, identifying the enhancer location (Fig 3A). Enhancer transcription at this location was absent in the chronic myelogenous leukemia cell line K562, which was consistent with lower transcription of *TAL1* compared with Jurkat cells.

Elevated expression of the oncogenic transcription factor c-MYC (hereafter referred to as MYC) is a hallmark of cancer (Gabay *et al*, 2014) and is largely established by the formation of numerous tumor-specific enhancers in the gene desert surrounding *MYC* (Ahmadiyeh *et al*, 2010; Chapuy *et al*, 2013; Hnisz *et al*, 2013; Herranz *et al*, 2014; Fulco *et al*, 2016; Zhang *et al*, 2016; Dave *et al*, 2017). TT-seq unveiled the extent and specificity of enhancer transcription across multiple diverse cancer cells, recapitulating the *MYC* gene's complex *cis*-regulatory architecture in great detail (Fig 3B). For the majority of cell lines, this represents the first map of enhancer transcription at the *MYC* locus. Importantly, enhancer transcription was present in regions enriched in genetic variants predisposing to cancer (Fig 3B), previously

annotated enhancers based on histone modifications and chromatin accessibility (Hnisz *et al*, 2013; Corces *et al*, 2018), and functionally verified enhancers (Fig EV2A, Appendix Fig S3) (Fulco *et al*, 2016; Dave *et al*, 2017).

To further illustrate that TT-seq reliably measures enhancer transcription at functionally verified enhancers, we used CRISPR/Cas9-mediated deletion of a 520 kbp enhancer region upstream of the *MYC* gene to extend published results (Dave *et al*, 2017). *Dave et al* showed that deletion of this enhancer region decreases *MYC* expression in multiple mouse tissues and impairs cell proliferation in the colorectal cancer cell line GP5d. The presence of active enhancers in this region agrees well with the strong eRNA synthesis we measured in GP5d cells (Figs 3B and EV2A). In contrast, we observed almost no eRNA synthesis across this region in the prostate cancer cell line DU 145 (Figs 3B and EV2A), suggesting little or no enhancer activity in DU 145 cells. When we carried out CRISPR/Cas9-mediated deletion of this region in DU 145 cells, edited cells were not depleted from the cell population during co-culture with unedited cells (Fig EV2B), whereas this was observed in GP5d cells (Dave *et al*, 2017). This indicates that the deleted region is not necessary for cell proliferation in DU 145 cells and might not contain an active enhancer element regulating *MYC* expression. In summary, our data capture precise maps of enhancer transcription at well-studied and functionally verified enhancers, and thus, TT-seq is a well-suited method to measure enhancer transcription.

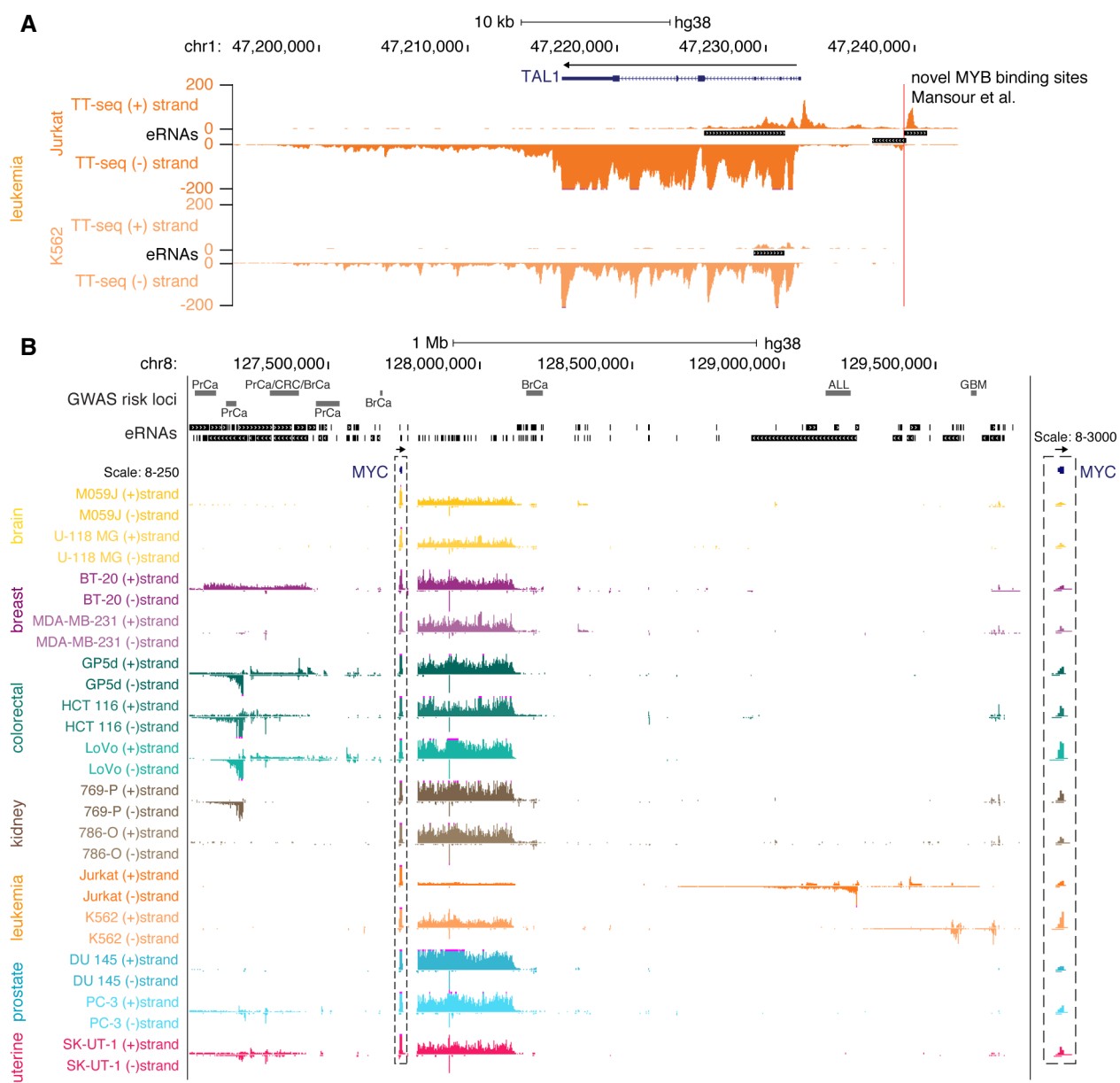

**Figure 3. Transcription at functionally verified enhancers.**

A UCSC genome browser view of normalized TT-seq coverage on the plus and minus strand at the *TAL1* locus (hg38; chr1:47,194,268–47,242,880 (Kent *et al*, 2002)) in the leukemia cell lines Jurkat and K562. For better visualization, TT-seq coverage is cut at 200 (purple lines). eRNAs are shown between the plus and minus strand. Red line indicates the short 12-bp insertion in Jurkat cells, introducing novel binding sites for the transcription factor MYB (Mansour *et al*, 2014).

B UCSC genome browser view of normalized TT-seq coverage on the plus and minus strand at the 2.8 Mbp gene desert surrounding *MYC* (hg38; chr8:127,042,406–129,779,109 (Kent *et al*, 2002)) in fourteen cancer cell lines. eRNAs represent the union from all cell lines. Genome-wide association studies (GWAS) risk loci contain single nucleotide polymorphisms (SNPs) for prostate cancer (PrCa), acute lymphocytic leukemia (ALL), breast cancer (BrCa), colorectal cancer (CRC), and glioblastoma (GBM), including SNPs in linkage disequilibrium ($r^2 \geq 0.8$). TT-seq coverage is displayed between 8 and 250 to allow for better visualization of the eRNA signal surrounding *MYC*. Dashed rectangle is shown at full scale on the right. Note: TT-seq reads were mapped against the hg38 reference genome, and we cannot rule out the existence of genomic rearrangements (e.g., translocations) affecting the shown region in the used cancer cell line clones.

### Comparison to high-throughput functional assays

To further evaluate the functionality of TT-seq-defined putative enhancers, we examined their activities in genome-wide plasmid-based reporter assays. To this end, we used HCT 116 and K562 STARR-seq data available from the ENCODE portal and investigated the overlap between TT-seq-defined enhancers and STARR-seq peaks (Materials and Methods). About 14 and 10% of TT-seq-defined enhancers were active in STARR-seq for HCT 116 and K562, respectively (Fig EV3A). Next, we compared these results to STARR-seq overlaps of putative enhancers called by other methods and sets of randomly sampled genomic regions (Materials and

Methods). The proportions of TT-seq enhancers with STARR-seq activity were larger than the proportions observed for open chromatin (DHS) or H3K4me1-defined enhancers, similar to H3K27ac-defined enhancers, and more than 6-fold enriched over background (Fig EV3A). We further show that taking a conservative subset of putative enhancers by combining DHS, H3K27ac and TT-seq, increased the STARR-seq overlap to 18 and 14% for HCT 116 and K562, respectively (Fig EV3A). We hypothesized that some enhancers are not active in STARR-seq as they might be part of cooperating enhancer clusters and thus might not function when tested individually. To investigate this, we counted the number of neighboring enhancers within ± 12.5 kbp for each putative enhancer and found that enhancers not active in STARR-seq were significantly enriched for enhancers with neighboring enhancers located in close proximity (Fig EV3B).

To extend these results, we asked whether ENCODE-predicted enhancers with validated functionality in K562 cells, as determined by CRISPRi screens (Gasperini *et al*, 2019; Schraivogel *et al*, 2020), were enriched for TT-seq-defined enhancers. Indeed, enhancers tested in either of the two studies were more likely to be functional if they overlapped a TT-seq-defined enhancer (Fisher's exact test, *P*-values < 0.01). Finally, we took the verified functional enhancer regions from both studies and calculated the overlap with the K562 STARR-seq peaks. The proportion of CRISPRi-verified functional enhancers also called active by STARR-seq was 25 and 14% for *Gasperini et al* and *Schraivogel et al*, respectively. Together these results corroborate the putative enhancers defined by TT-seq in our study.

### eRNA synthesis at super-enhancers and in large regulatory regions

We next investigated the relationship between our eRNA-defined enhancers and super-enhancers. Super-enhancers are defined as large clusters of enhancers that are densely occupied by master transcription factors and the Mediator complex at key cell identity genes (Parker *et al*, 2013; Whyte *et al*, 2013). Using Med1 ChIP-seq data (Yan *et al*, 2013), we identified ~ 3,500 typical and 214 super-enhancers in LoVo colorectal cancer cells (Fig 4A, Materials and Methods). We found that super-enhancers were more highly transcribed than typical enhancers (Fig 4B), consistent with previous results (Hnisz *et al*, 2013; Hirabayashi *et al*, 2019). Moreover, eRNAs originating from super-enhancers were significantly longer compared with other eRNAs (Fig 4C). About 90% of super-enhancers contained at least one transcribed enhancer region as defined by TT-seq, with most super-enhancers comprising one to three enhancers, and some super-enhancers covering up to seven enhancers (Fig 4D).

Conversely, we found large regulatory regions of the order of several hundreds of kilobase pairs characterized by several super-enhancers, and densely clustered and strongly transcribed individual enhancers, e.g., next to *PHLDA1* (Fig 4E), *MYC* (Figs 3B and EV4A), and *KLF5* (Fig EV4B). The 500 kbp region downstream of *PHLDA1* encompassed 41 enhancer regions in HCT 116 colorectal cancer cells of which 35 overlapped with four super-enhancers (Jiang *et al*, 2019). These enhancers produced a combined transcript length of ~ 400 kb, correlated with *PHLDA1* transcription, and are likely relevant for *PHLDA1* gene expression because disruption of

individual enhancers in this region reduces *PHLDA1* expression in HCT 116 cells (Cohen *et al*, 2017). Each individual enhancer was bound by combinations of transcription factors and was co-occupied with the histone acetyltransferase p300 and Pol II (Fig 4E). This dense clustering of transcription factors supported the classification of these extensive transcripts as eRNAs originating from clustered enhancers. Among the transcription factors binding to almost all individual enhancers were YY1, a largely pro-tumorigenic transcription factor in colon cancer (Sarvagalla *et al*, 2019), SP1, which has a significant role in colorectal cancer development and progression (Bajpai & Nagaraju, 2017), and TCF7l2, which plays a key role in the Wnt signaling pathway and is activated in most colorectal cancers (Bienz & Clevers, 2000) (Fig 4E).

### Cell type specificity of enhancer transcription

To compare the transcriptional landscape at promoters and enhancers across cell lines, we first used hierarchical clustering of TT-seq signals from mRNAs and eRNAs. This grouped the samples, independently of gender, first by replicates and then by tissue/cancer type, except for breast cancer samples, which were slightly more distinct from each other (Fig 5A, Appendix Fig S4A). For mRNA synthesis, we observed a high correlation across cell lines, whereas eRNA synthesis was less correlated (Fig 5A). Importantly, the variability of enhancer transcription seemed to follow a controlled transcriptional program, as the biological replicates were highly correlated (Appendix Fig S4A) and the tissue/cancer type identity of cell lines was mostly maintained (Fig 5A).

The higher variability of eRNA transcript levels could be due to enhancers that are active in only a limited number of cell and tissue types or due to the strongly varying activity of ubiquitously transcribed (i.e., transcribed in all cell lines) enhancers, or both. Using the binary information of mRNA and eRNA being synthesized versus not synthesized, we found that 58.5% of mRNAs were ubiquitously synthesized in all cell lines, and only 4.8% were exclusively synthesized in a single cell line (Fig 5B). This was in contrast to eRNAs of which only 2.1% were ubiquitously synthesized and more than one third (37.2%) was exclusively synthesized in a single cell line (Fig 5B).

Although a large fraction of mRNAs was synthesized ubiquitously, their synthesis level across cell lines often varied (Appendix Fig S4B). Thus, to examine cell type-specific mRNA and eRNA synthesis in more detail, we calculated a cell type specificity score for each mRNA and eRNA using an entropy-based metric (Materials and Methods). This specificity score was defined to range between 0 and 1, where 0 means unspecific (i.e., uniform synthesis level across all cell lines) and 1 means specific (i.e., specific synthesis level in one cell line). These entropy-based specificity scores also showed that eRNAs were synthesized in a more cell type-specific manner than mRNAs (Fig 5C). Together these results befit the known cell type-specific activity of enhancers (Voss *et al*, 1986; Heinz *et al*, 2015).

### Cell type-specific enhancers are associated with cell type-specific gene activity

These results beg the question how the specific enhancer transcription can coincide with the less specific gene transcription from

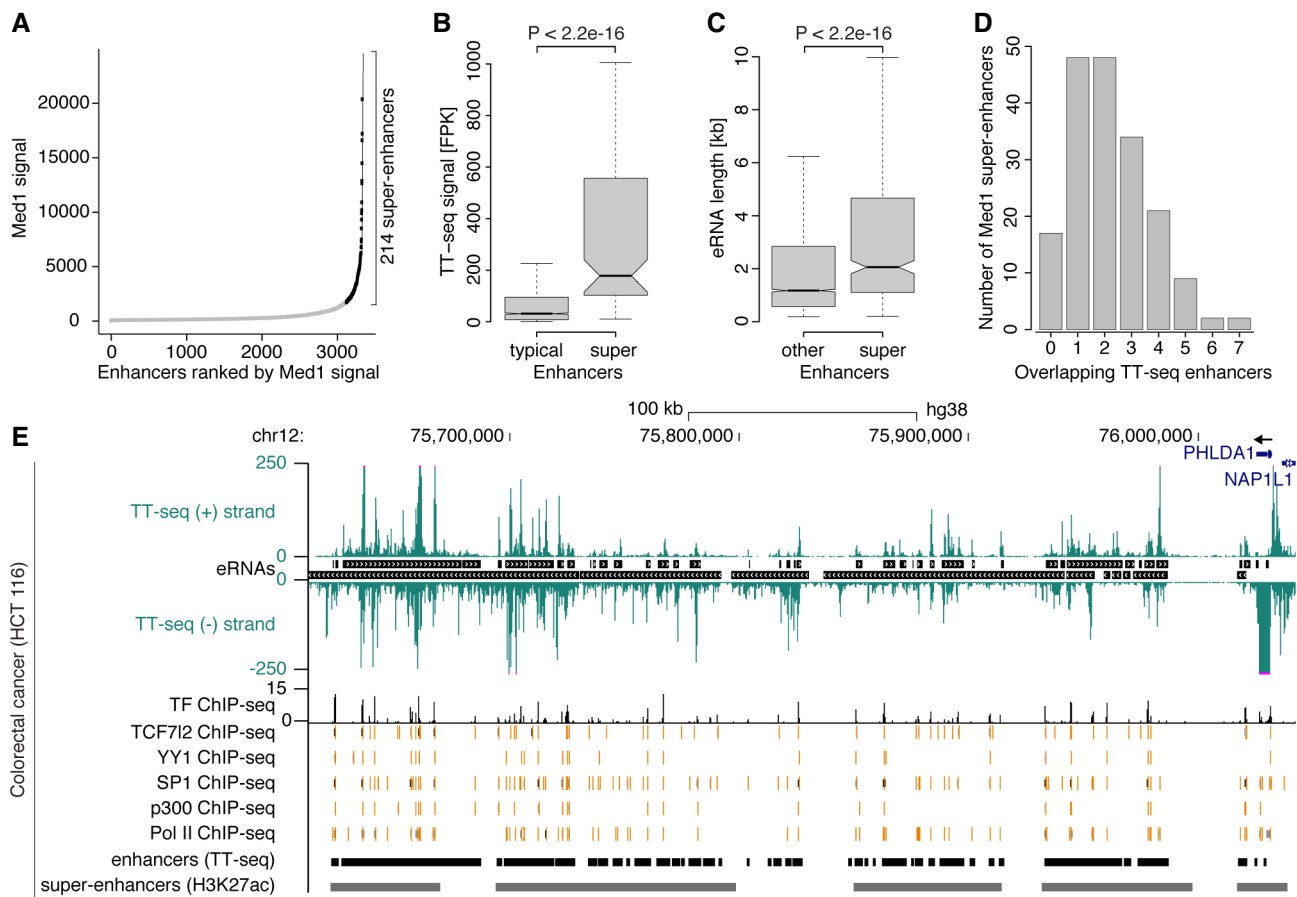

**Figure 4. eRNA synthesis at super-enhancers and in large regulatory regions.**

A  Med1-defined enhancers ranked by Med1 ChIP-seq signal (Yan *et al*, 2013) in LoVo colorectal cancer cells. Super-enhancers (black rectangles) were determined as enhancer regions lying above the inflection point of the curve (Materials and Methods).

B  Increased eRNA synthesis from super-enhancers. Boxplots comparing eRNA synthesis (TT-seq signal) at Med1-defined typical enhancers (*n* = 2,111) and super-enhancers (*n* = 130) in LoVo cells (analysis was restricted to intergenic enhancers; Materials and Methods). Box limits are the first and third quartiles, and the band inside the box is the median. The ends of the whiskers extend the box by 1.5 times the interquartile range. Notches represent 95% confidence intervals for the median values. Outliers not shown. *P*-value by two-sided Mann–Whitney *U*-test.

C  Super-enhancers are associated with the production of long eRNAs. Boxplots showing length of TT-seq-defined eRNAs originating from Med1-defined super-enhancers (*n* = 552) or other enhancer regions (*n* = 6,934). Median, hinges, whiskers, and notches are shown as in (B). *P*-value by two-sided Mann–Whitney *U*-test.

D  Barplot showing the number of TT-seq enhancer regions per Med1-defined super-enhancer region in LoVo cells. Super-enhancers spanning a protein-coding TSS (*n* = 32) are excluded from this comparison as regions TSS ± 1 kbp were excluded from TT-seq enhancer calling.

E  UCSC genome browser view of normalized TT-seq coverage on the plus and minus strand at the *PHLDA1* locus (hg38; chr12:75,612,350–76,042,552 (Kent *et al*, 2002)) in the colorectal cancer cell line HCT 116. TT-seq coverage is cut at 250 (purple lines) to allow for better visualization. eRNAs are shown between the plus and minus strand. Aggregated transcription factor ChIP-seq peak regions (TF ChIP-seq) comprise ChIP-seq binding profiles of 19 TFs in HCT 116 cells from CISTROME (Zheng *et al*, 2019). The height of the signal indicates the number of different TFs binding at a particular region. Binding regions for selected TFs (TCF7l2, YY1, SP1) are shown below together with p300 and RNA polymerase II (Pol II). TT-seq enhancer regions are shown at the bottom together with a H3K27ac-defined super-enhancer annotation, which was downloaded from SEdb (Jiang *et al*, 2019). At date of publication, no Med1 ChIP-seq data were available for HCT 116.

promoters. To investigate this, we paired enhancers to their nearest active promoter (Fig 6A; "nearest" approach). We then filtered for enhancers that were exclusively active in a single cell line and determined the transcription level for each enhancer (Appendix Fig S4C) and its nearest active promoter (Fig 5D) over all cell lines. Nearest active promoters showed higher transcription signal in the cell line in which the enhancer was transcriptionally active as compared to all other cell lines, indicating that transcriptionally active enhancers are associated with cell type-specific promoter activity (Fig 5D). Yet, in only 3.6% of cases was transcription of the nearest active

promoter exclusively observed in the cell line in which the enhancer was exclusively active, and in more than half of the cases, nearest promoter transcription was observed in all cell lines (Fig 5E). Results were overall similar when instead of analyzing nearest promoters we selected promoters with observed physical proximity to the exclusively transcribed enhancers (Appendix Fig S4D and E).

To extend these results beyond enhancers that are active in a single cell line, we correlated the entropy-based cell type specificity scores of enhancer and promoter transcription. When specificity scores of enhancers increased, the specificity scores of the nearest

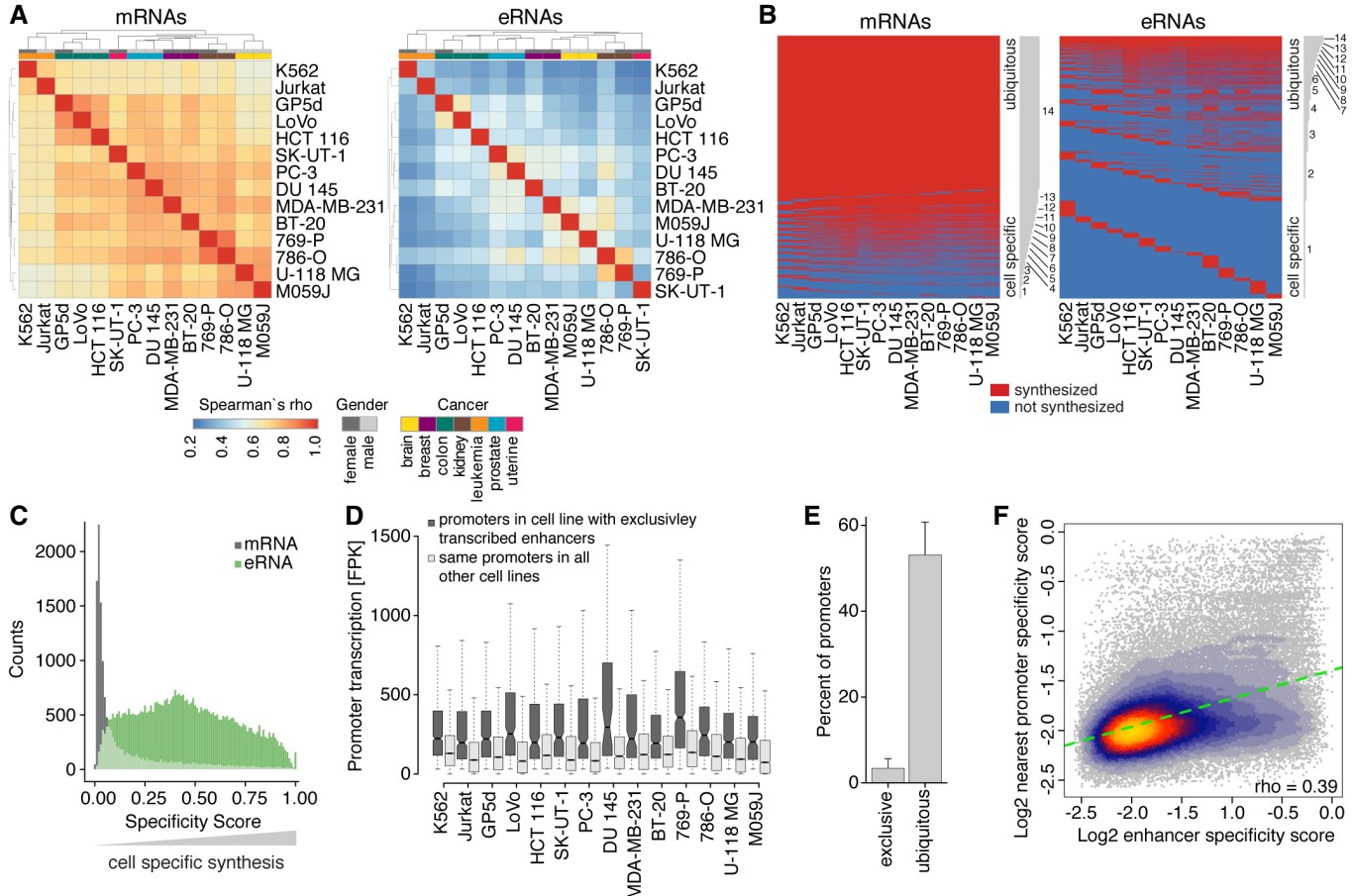

**Figure 5. Cell type specificity of enhancer transcription.**

A   Unsupervised clustering of all pairwise Spearman correlations of normalized TT-seq signal for mRNAs (left) and eRNAs (right) for the union of transcripts over all fourteen cancer cell lines (Materials and Methods). Gray and colored bars on top indicate gender and cancer type, respectively.

B   Heatmaps showing mRNAs (left) and eRNAs (right) as being synthesized (red, normalized TT-seq signal FPK ≥ 15) or not (blue, normalized TT-seq signal FPK < 15) for all fourteen cancer cell lines (columns). Heatmaps show only transcripts (rows) with normalized FPK ≥ 30 in at least one of the cell lines (mRNAs, $n = 14,040$; eRNAs $n = 48,105$). Transcripts are ordered by increasing cell type specificity, as indicated by the gray bars to the right showing the number of cell lines a transcript is observed in.

C   Distribution of entropy-based cell type specificity scores for mRNAs (gray) and eRNAs (green) as in (B). The scores range between 0 and 1, where 0 means unspecific (i.e., uniform synthesis level across all cell lines) and 1 means specific (i.e., specific synthesis level in one cell line).

D   Boxplots showing normalized TT-seq signal for nearest active promoters in the cell line in which enhancers are exclusively transcribed (from left to right: $n = 890$; 500; 494; 214; 238; 239; 378; 62; 153; 879; 586; 173; 601; 232) compared with the same promoters in all other cell lines in which the respective enhancers are not transcribed. All pairwise comparisons are significantly different ($P$-values < 1e-9, two-sided Mann–Whitney $U$-test). Box limits are the first and third quartiles, and the band inside the box is the median. The ends of the whiskers extend the box by 1.5 times the interquartile range. Notches represent 95% confidence intervals for the median values. Outliers not shown.

E   Barplot showing percentage of nearest active promoters for which transcription is observed exclusively in the same cell line as the respective exclusively in this cell line transcribed enhancers (exclusive) or is observed in all fourteen cell lines (ubiquitous). Percentages are expressed as mean over all 14 cell lines ± SD.

F   Heat scatterplot comparing log$_2$ cell type specificity scores for enhancers ($x$-axis) and nearest active promoters ($y$-axis) over all cell lines. Spearman correlation = 0.39. Dashed green line denotes linear regression line with slope = 0.28.

promoters also increased (Fig 5F). Together, these results indicate that transcribed putative enhancers identified using TT-seq data are associated with cell type-specific gene activity.

**Enhancer–promoter pairing based on transcription correlation**

Computationally linking enhancers to their target genes remains challenging and multiple approaches beyond pairing enhancers to the nearest active gene exist (Hariprakash & Ferrari, 2019).

Benefiting from our rich data collection across multiple cell lines, we made use of pairwise correlation between transcriptional activity at gene promoters and enhancers to define putative enhancer–gene promoter (E–P) pairs that show highly correlated transcription across cell lines. We further combined the correlation-based pairing strategy with two different distance constraints (Fig 6A). In the "correlated neighboring" (CN) approach, we pair enhancers with the nearest upstream and downstream neighboring promoters. In the "correlated window" (CW) approach, we pair enhancers to all

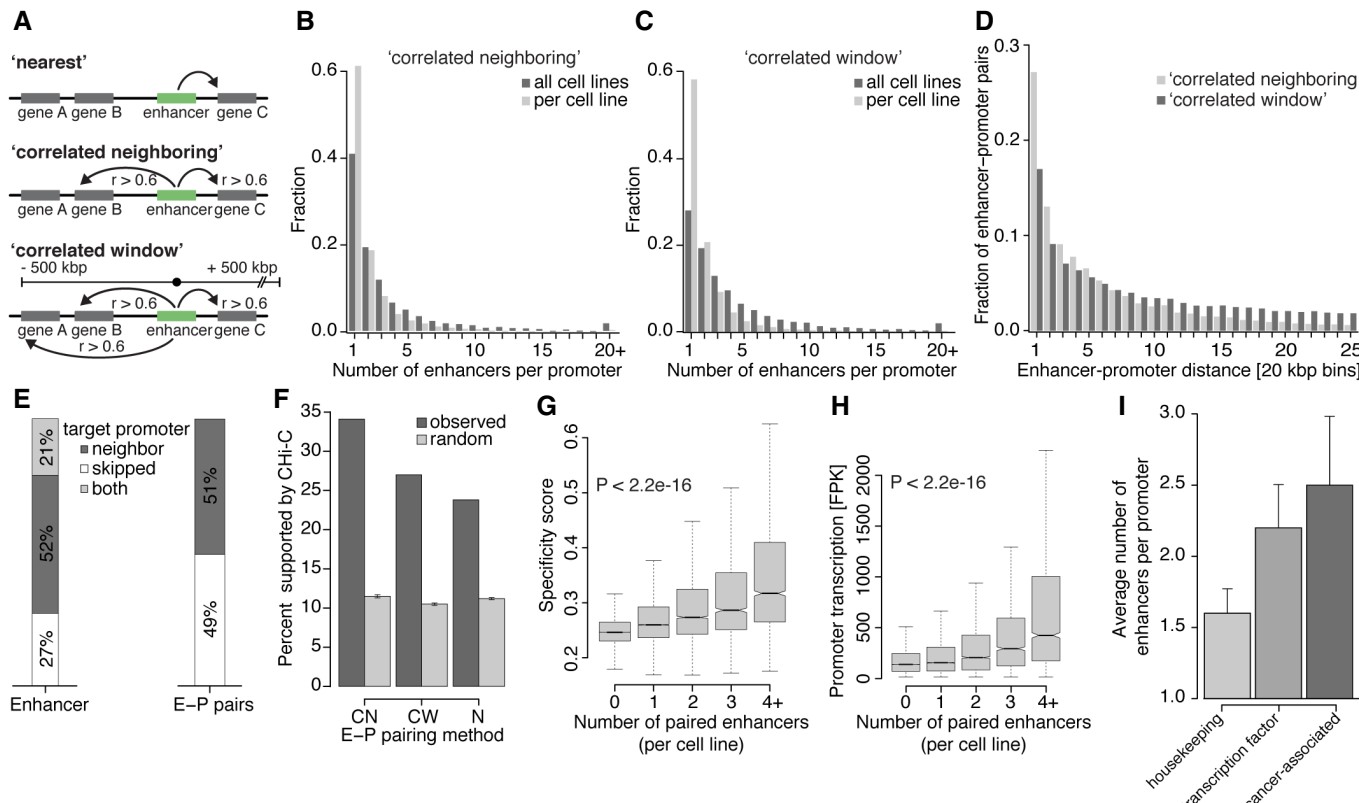

**Figure 6. Enhancer–promoter pairing based on transcription correlation.**

A Schematic diagram describing enhancer–promoter pairing methods. With the "nearest" (N, top), "correlated neighboring" (CN, middle), and "correlated window" (CW, bottom) approach, enhancers are paired to the nearest transcribed promoter, the neighboring upstream and downstream transcribed promoters, and all transcribed promoters within ± 500 kbp distance, respectively. For CN and CW only pairs with correlated transcription rates over all cell lines (Pearson correlation coefficient *r* > 0.6) were analyzed further (Materials and Methods).

B Number of enhancers per promoter, for all promoters that were paired to at least one enhancer with the CN approach. Fractions are shown for enhancers per promoter per cell line (light gray) and over all cell lines (dark gray).

C As in (B) for pairs obtained with the CW approach.

D Distance distribution between enhancer midpoints and promoter TSSs for E–P pairs obtained with the CN (light gray) and CW (dark gray) approach. The histogram shows the 500 kbp distance range in 20 kbp bins.

E Left: stacked barplot showing the fraction of enhancers that pair to neighboring transcribed promoters (neighbor, dark gray), to transcribed promoters further away thereby skipping the transcribed neighboring promoters (skipped, white), or both (light gray) using the CW approach. Right: stacked barplot showing the fraction of E–P pairs between enhancers and their neighboring transcribed promoters (dark gray) as well as between enhancers and promoters skipping a transcribed neighboring promoter (white).

F Barplot showing the percentage of E–P pairs supported by promoter capture Hi-C (CHi-C) interactions for different pairing approaches (dark gray). Expected interactions based on 100 sets of randomly sampled matched E–P pairs (light gray, Materials and Methods). Light gray bars depict the mean, and error bars represent 95% confidence intervals.

G Boxplots showing promoter cell type specificity score for different numbers of paired enhancers using the CN approach (from left to right: *n* = 124,767; 12,347; 3,754; 1,619; 2,373). Indicated *P*-value by Kruskal–Wallis test. All pairwise comparisons are significantly different (*P*-values < 1e-10, two-sided Mann–Whitney *U*-test). Specificity scores and number of enhancers were assessed for each cell line (Materials and Methods). Box limits are the first and third quartiles, and the band inside the box is the median. The ends of the whiskers extend the box by 1.5 times the interquartile. Notches represent 95% confidence intervals for the median values. Outliers not shown.

H As in (G) showing promoter transcription. Indicated *P*-value by Kruskal–Wallis test. All pairwise comparisons are significantly different (*P*-values < 2.2e-16, two-sided Mann–Whitney *U*-test).

I Barplots showing the average number of paired enhancers per promoter per cell line (*n* = 14, mean ± SD, CN approach) for promoters paired to at least one enhancer and being classified as housekeeping genes ((Eisenberg & Levanon, 2013) left), transcription factors ((Abugessaisa *et al*, 2016) middle), or cancer-associated genes (Cancer Gene Census (Sondka *et al*, 2018) right).

transcribed promoters that lie within a ± 500 kbp distance. For both approaches, we only kept E–P pairs if the Pearson correlation coefficient between synthesis of eRNA and mRNA was > 0.6 (Materials and Methods).

For the CN approach, we obtained a total of 21,076 putative E–P pairs (Table EV2). These pairs contained 18,828 transcribed promoters and 6,028 promoters, representing 32% and 43% of all enhancers and active promoters, respectively. Among all E–P pairs, a promoter paired on average to 2.0 enhancers in a given cell line (Fig 6B). Notably, a promoter paired on average to 3.5 enhancers when we considered the union of paired enhancers over all cell lines, indicating that the enhancers regulating the same promoter

can vary between cell lines (Fig 6B). For the CW approach, a total of 41,211 putative E–P pairs (Table EV2) contained 25,719 transcribed enhancers (44%) and 9,534 promoters (68%). Here, a promoter was paired on average to 2.0 and 4.3 enhancers per cell line and over all cell lines, respectively (Fig 6C). Whereas the CN approach limits the number of enhancers paired to each promoter to a maximum of two, the CW approach allows pairing of multiple promoters within a window of ± 500 kbp. Yet, only 12% of enhancers paired to more than two promoters and the paired promoters were generally the same in different cell lines (Appendix Fig S5A).

Next, we compared E–P distances for both correlation-based approaches. Median distances were 62 kbp and 122 kbp, with 45% and 30% of all paired enhancers residing within ± 50 kbp from their paired promoter for the CN and CW approach, respectively (Fig 6D). Interestingly, the CW approach revealed that 52% of enhancers paired exclusively to one of their active neighboring promoters and only 27% of enhancers skipped their two neighboring promoters. 21% of enhancers paired to their neighboring promoters and additionally to promoters further away (Fig 6E). Considering the total of 40,772 E–P pairs, 51% occurred between enhancers and their neighboring promoters, and in the other 49% of pairs, the neighboring promoters were skipped (Fig 6E). Notably, the correlation of transcription within E–P pairs decreased with increasing distance between enhancer and promoter (Appendix Fig S5B), marking close E–P pairs as more confident.

To further evaluate how the identified E–P pairs agree with physical proximity as measured by the global chromosome conformation mapping technique Hi-C, we first used available promoter capture Hi-C (CHi-C) data for 19,023 promoter fragments measured in LoVo colorectal cancer cells (Orlando *et al*, 2018). To assess enrichment of observed CHi-C interactions, we generated 100 matched sets of random E–P pairs; i.e., promoters were chosen to match promoter activity and cell type specificity, and non-coding regions were chosen to match enhancer width and distance (Materials and Methods). E–P pairs in LoVo cells were significantly enriched for CHi-C interactions, and the percentages of pairs with CHi-C support were 23.8% ("nearest"), 34.1% ("correlated neighboring"), and 27% ("correlated window") (Fig 6F). Moreover, E–P pairs involving neighboring promoters were enriched for CHi-C interactions compared with E–P pairs skipping at least one neighboring TSS (Fisher's exact test, odds ratio = 2.0, *P*-value < 2.2e-16). Next, we compared our E–P pairing approach with the activity-by-contact (ABC) model (Fulco *et al*, 2019), which predicts E–P interactions based on Hi-C contact frequency between enhancer and promoter, and DHS and H3K27ac signal as proxy for enhancer activity. We selected all putative enhancers that were paired to at least one promoter by both our strategy and the ABC model, and compared the overlap between the predicted E–P connections in K562 cells. Using this subset of enhancers, we found that the majority (64%) of the putative E–P pairs reported here were also predicted by the ABC model.

Finally, we quantified the effect of multiple enhancers per promoter and observed that both cell type specificity and level of promoter transcription increased with the number of paired enhancers (Fig 6G and H, Appendix Fig S5C–F). In addition, the number of enhancers was also related to gene function (Fig 6I). Whereas housekeeping genes were underrepresented, paired promoters were enriched for encoding transcription factors and cancer-associated

genes (Fisher's exact test, odds ratios = 0.4, 1.58, 1.53, respectively, *P*-values < 2.2e-16).

## Cancer-associated DNA sequence variation in transcribed enhancers

In total, our 58,457 transcribed putative enhancers collectively span 3.6% of the human genome. To investigate the extent to which these enhancers overlap with cancer-associated single nucleotide polymorphisms (SNPs), we compiled a list of 2,459 distinct SNPs from the NHGRI-EBI genome-wide association studies (GWAS) catalog (Welter *et al*, 2014) that are linked to the seven cancer types we studied (Materials and Methods). We found that 94% of these cancer-associated SNPs occur in non-coding regions and that 8.3% of these occur within transcribed enhancer regions. Thus, 8.3% of cancer-associated non-coding SNPs occur in 3.6% of the genome. To further confirm this enrichment, we randomly sampled 1,000 SNP sets from the 1000 Genomes Project (Auton *et al*, 2015) across the non-coding genome, matching distance to nearest gene, gene density, SNPs in linkage disequilibrium (LD), and allele frequency using SNPsnap ((Pers *et al*, 2014) Materials and Methods). Transcribed enhancers had significantly more overlaps (168) with GWAS cancer-associated SNPs than with any of the matched SNP sets (median = 87) (Fig EV5, *P*-value < 0.001, simulation). Further, enhancers were more enriched for cancer-associated SNPs than all other SNP-disease/trait associations from GWAS (Fisher's exact test, odds ratio = 1.53, P-value = 1.183e-7). Finally, we selected only enhancers paired to at least one gene by any of the three pairing strategies (Fig 6A) and divided them into two groups depending on if they paired to a cancer gene from the Cancer Gene Census (Sondka *et al*, 2018), or not. We found that cancer-associated SNPs were enriched in enhancers paired to cancer genes (Fisher's exact test, odds ratio = 1.69, P-value = 0.01318). Together, these results confirm the prevailing occurrence of genetic risk variants in enhancers (Sur & Taipale, 2016) and strengthen our annotation of transcribed putative enhancers in cancer cell lines as a valuable resource for studying somatic mutations in the non-coding genome.

## A resource of transcribed enhancers in human cancer cell lines

To aid studies of cancer genomics, we created a comprehensive catalog of transcribed putative enhancers with cancer-associated somatic mutations (Table EV3) and putative enhancer–promoter pairs involving cancer genes from the Cancer Gene Census (Sondka *et al*, 2018) (Table EV4). In total, we derived 5,296 E–P pairs including 5,154 transcribed putative enhancers and 593 cancer genes, covering 82% of all Cancer Gene Census genes. In addition, putative E–P pairs involving cancer-related genes not included in the Cancer Gene Census can be retrieved from our general E–P list (Table EV2). One example is the oncogene encoding the bone morphogenetic protein 4 (BMP4), a member of the transforming growth factor-ß family that is universally upregulated in human colorectal cancer cells and tissues (Yokoyama *et al*, 2017). Based on all three E–P pairing approaches, *BMP4* is potentially regulated by multiple enhancers located within a ~ 300 kbp region upstream of *BMP4* in LoVo colorectal cancer cells. These enhancers were characterized by particularly strong eRNA synthesis, ChIP-seq binding peaks of combinations of transcription factors (Yan *et al*, 2013), and chromatin interactions between the *BMP4* promoter and TT-seq

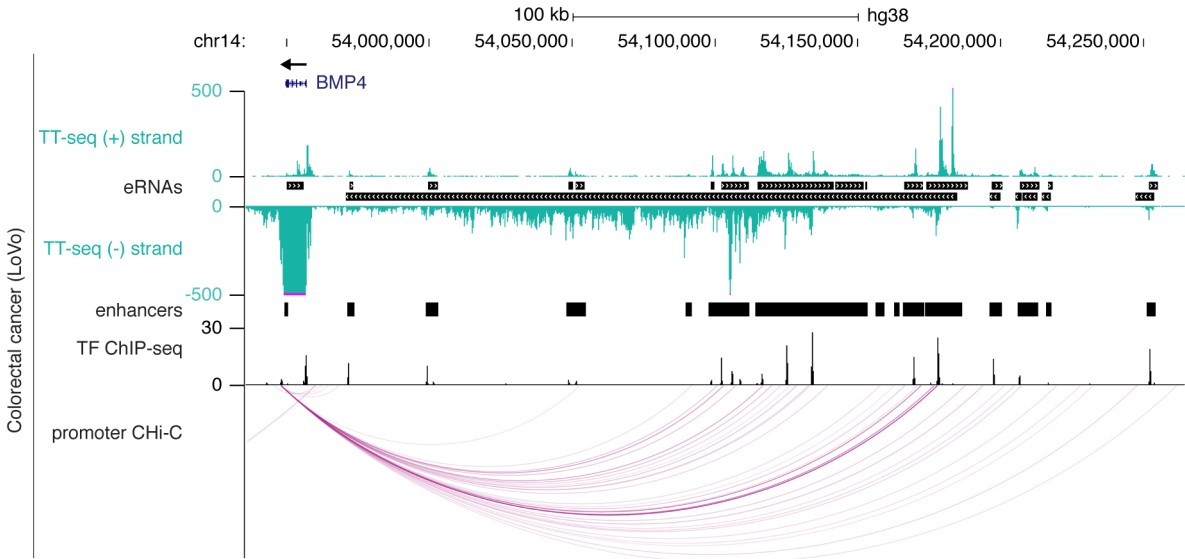

**Figure 7. Putative enhancers regulating the oncogene *BMP4*.**

UCSC genome browser view of normalized TT-seq coverage on the plus and minus strand at the *BMP4* locus (hg38; chr14:53,935,697–54,264,430 (Kent *et al*, 2002)) in LoVo colorectal cancer cells. TT-seq coverage is cut at 500 (purple lines) to allow for better visualization. eRNAs and enhancers are shown above and below the minus strand, respectively. Aggregated transcription factor ChIP-seq peak regions (TF ChIP-seq) include ChIP-seq binding profiles of 326 TFs in LoVo cells (Yan *et al*, 2013). The height of the signal indicates the number of different TFs binding at a particular region. Significant promoter capture Hi-C (CHi-C) interactions are shown in purple (Orlando *et al*, 2018). See Source Data for Fig 7 for original view of CHi-C interactions from WashU epigenome browser (Li *et al*, 2019).

Source data are available online for this figure.

annotated enhancers, as revealed by promoter CHi-C data (Orlando *et al*, 2018) (Fig 7).

The strength and ease of use of the resource presented here is that (i) it is a simultaneous and quantitative analysis of promoter and enhancer transcription, (ii) it provides TT-seq signal tracks together with putative enhancer and eRNA annotations, (iii) it provides putative enhancer–promoter pairs based on several pairing approaches, and (iv) by using enhancer transcription it increases the fidelity of selecting functional enhancers for further analyses. We anticipate many applications of this resource to further study enhancer activity and gene regulation. The resource will also help to study how somatic mutations in enhancers and transcription-related factors alter enhancer function and specificity to cause a cancer phenotype.

## Discussion

A major challenge in understanding the role of enhancers in cancer is the accurate identification of active enhancer elements and uncovering the genes they regulate. Here, we used TT-seq to measure eRNA synthesis and derived a refined approach to annotate transcribed putative enhancers. This resulted in a large collection of *in vivo* transcribed candidate enhancers in fourteen human cancer cell lines from seven different types of cancer. The putative enhancers are in accordance with known genomic marks of active enhancers (Figs 1 and EV1, Appendix Fig S1), functionally verified enhancers (Figs 3, 4E, EV2 and EV3, Appendix Fig S3), cell type-specific patterns of gene expression (Fig 5, Appendix Fig S4), and are enriched for genetic variants that predispose to cancer (Figs 3B

and EV5). Our resource of transcribed putative enhancers expands existing enhancer transcription data to other cancer types and cell lines, and complements previous mappings of putative enhancers in cancer cells that used different techniques such as RNA-seq and CAGE (Chen *et al*, 2018), ChRO-seq (Chu *et al*, 2018), ATAC-seq (Corces *et al*, 2018), PRO-seq (Danko *et al*, 2015), GRO-seq (Franco *et al*, 2018), NET-CAGE (Hirabayashi *et al*, 2019), and mapping of the H3K27ac mark (Hnisz *et al*, 2013; Cohen *et al*, 2017).

To further assess the functionality of the putative enhancers described here, we compared them to available data from two types of large-scale functional assays: genome-wide plasmid-based reporter gene assays (STARR-seq (Davis *et al*, 2018)) and high-throughput CRISPRi screens (Gasperini *et al*, 2019; Schraivogel *et al*, 2020). In HCT 116 and K562 cells, 14% and 10% of TT-seq-defined enhancer regions overlapped a region with STARR-seq activity. Overlaps increased to 18% and 14%, respectively, when we used a more constrained set of candidate enhancers, which were annotated by several enhancer marks (DHS, H3K27ac, and TT-seq). Overall, this represents a good overlap since even for CRISPRi-validated enhancers in K562 (Gasperini *et al*, 2019; Schraivogel *et al*, 2020) less than one-fourth (162/677) were active in STARR-seq. Many possible reasons exist for why a CRISPRi-validated enhancer would fail to show activity in a reporter gene assay, in which enhancers are tested outside of their endogenous chromatin environment. One could be that some of the enhancers are part of interacting enhancer clusters and might show no or low reporter gene activity when tested individually. In agreement with this, we found that TT-seq-defined putative enhancers not active in STARR-seq were enriched for enhancers with neighboring enhancers located in close proximity (Fig EV3B).

Accurate identification of target genes regulated by enhancers remains challenging. Several computational approaches have been proposed (reviewed in (Hariprakash & Ferrari, 2019)), and their applicability depends on the genomic data sets that are available. Since there are enhancers known to regulate genes over long distances, even spanning regions in the Mb range, and/or to skip over neighboring genes, the target space to look for putative target promoters of an enhancer becomes very large. Pairing enhancers to every gene within a certain distance (e.g., here, ± 500 kbp) would result in a high fraction of false-positive predictions (Fulco *et al*, 2019) and strategies to decrease the number of false-positive E–P pairs include considering only pairs with observed physical proximity (Javierre *et al*, 2016; Fulco *et al*, 2019) or with correlated activity (Thurman *et al*, 2012; Andersson *et al*, 2014). The fraction of physically interacting E–P pairs with correlated transcriptional activity was reported to be about 50% (Fitz *et al*, 2020). We decided to use a correlation cutoff for all putative E–P pairs that paired beyond the nearest active gene ("correlated window" approach), as physical interaction data were not available for most of the cell lines we investigated. Moreover, the strength of our approach is that TT-seq simultaneously quantifies the transcriptional activity of both gene promoters and enhancers. This allowed us to use correlation of transcriptional activity between enhancers and promoters across all samples to infer tens of thousands of putatively functional E–P pairs in an unbiased way (Fig 6). The pairs we derived are depleted for housekeeping genes and enriched for transcription factors, cancer-associated genes, and 3D conformational proximity (Fig 6). Finally, we provide Table EV5, which contains all possible E–P pairs within ± 500 kbp of each putative enhancer together with the observed distances and correlations of transcriptional activity, providing flexibility to adjust distance and/or correlation constraints.

Our comprehensive annotation of putative enhancers and E–P pairs in human cancer cell lines provides a valuable tool for further studying enhancer function in cancer. Tumor-derived cell lines carry many of the original oncogenic alterations, and transcriptional profiles of these cell lines have been shown to be in good agreement with those of primary tumors (Greshock *et al*, 2007; Iorio *et al*, 2016; Ghandi *et al*, 2019). However, it is important to consider that cancer cell line models have major limitations and inevitably entail genomic differences as pointed out by several studies (Stein *et al*, 2004; Sandberg & Ernberg, 2005; Ertel *et al*, 2006; Gillet *et al*, 2011). Cell culturing conditions fail to recapitulate the condition of tumor cell growth in patients, and cell line models cannot accurately model the complexity of the patient tumor microenvironment, i.e., interaction with stroma and immune cells. Nevertheless, cancer cell lines are an indispensable tool in cancer research and preclinical studies of anticancer drugs (Sharma *et al*, 2010; Goodspeed *et al*, 2016; Katt *et al*, 2016; Mirabelli *et al*, 2019). Cancer cell lines can be readily manipulated, allow global and detailed mechanistic studies, and thereby provide insights into the molecular mechanisms of tumor growth. Several studies have characterized the genome-wide molecular profiles of cancer cell lines and investigated how well they model the genomic profiles, molecular subtypes, and heterogeneity of tumors (Holliday & Speirs, 2011; Barretina *et al*, 2012; Garnett *et al*, 2012; Domcke *et al*, 2013; Klijn *et al*, 2015; Ghandi *et al*, 2019). Together these results allow researchers to select cancer cell lines that best model the genomic features of a particular type of tumor. The large-scale whole-genome sequencing projects of cancer cell lines and cancer patient samples such as the Cancer Cell Line Encyclopedia (Ghandi *et al*, 2019) and the Pan-Cancer Analysis of Whole Genomes (Campbell *et al*, 2020) have constructed large catalogs of somatic mutations in many types of tumors. Integrating such data and extracting correlations between enhancer activity and tumor growth will be a valuable task for the future.

# Materials and Methods

### Cell culture

Human cancer cell lines were obtained from ATCC (Manassas, VA, USA) and Sigma-Aldrich (St. Louis, MO, USA), cultured in cell culture media (Gibco/Thermo Fisher Scientific, Waltham, MA, USA) supplemented with 1% penicillin/streptomycin (Cat No. 11556461; GE Healthcare, Chicago, IL, USA) and 10% fetal bovine serum (Cat No. F7524, Lot 035M3394; Sigma-Aldrich) at 37°C in a humidified 5% $CO_2$ incubator, and used for experiments at low passage numbers (< passage 14). BT-20 (ATCC HTB-19), DU 145 (ATCC HTB-81), and SK-UT-1 (ATCC HTB114) cells were cultured in MEM (Gibco, Cat No. 31095029) supplemented with 0.1 mM NEAA (Gibco, Cat No. 11140035) and 1 mM sodium pyruvate (Gibco, Cat No. 11360039). 769-P (ATCC CRL-1933) and 786-O (ATCC CRL-1932) cells were cultured in RPMI-1640 (Gibco, Cat No. A1049101). MDA-MB-231 (ATCC HTB-26) and LoVo (ATCC CCL-229) cells were cultured in DMEM-GlutaMAX (Gibco, Cat No. 21885-25). U-118 MG (ATCC HTB-15) and GP5d (Sigma, ECACC 95090715; RRID:CVCL_1235) cells were cultured in DMEM (Gibco, Cat No. 41966029). GP5d cells were cultured in cell culture flasks coated with poly-L-lysine (Cat No. P4707; Sigma-Aldrich). M059J (ATCC CRL-2366) cells were cultivated in DMEM/F-12 (Gibco, Cat No. 31330038), HCT 116 (ATCC CCL-247) cells were cultivated in McCoy's 5a Medium Modified (Gibco, Cat No. 26600023), and PC-3 (ATCC CRL-1435) cells were cultivated in F-12K (ATCC 30-2004). Cells were authenticated by STR profiling at the manufacturer (ATCC and ECACC) and verified to be free of mycoplasma contamination using the MycoAlertTM Mycoplasma Detection Kit (Cat No. LT07-318; Lonza, Basel, Switzerland). Biological replicates were cultured independently.

### TT-seq and RNA-seq

TT-seq experiments were performed in biological duplicates, and a full step-by-step protocol for TT-seq has been deposited in the protocols.io repository (Gressel *et al*, 2019a). In brief, 48 h before the experiment cells were seeded in 15-cm dishes to obtain 70% confluent cells at time of RNA labeling. The required number of cells depends on cell line-specific parameters (e.g., growth rate, cell size) and is listed in Appendix Table S1. Cells were labeled with 500 μM 4-thiouridine (4sU; Sigma-Aldrich, St. Louis, MO, USA) for 5 min at 37°C under 5% $CO_2$. Cells were harvested using TRIzol (Life Technologies/Thermo Fisher Scientific, Waltham, MA, USA). 2.4 ng RNA spike-ins mix per $10^6$ cells was added to each sample after cell lysis. Spike-in sequences and production are described in Gressel *et al* (2019a). Subsequently, total RNAs were extracted according to the TRIzol manufacturer's instructions. Total RNAs were sonicated to generate fragments of an average size < 15 knt (total fragmented

RNAs) using 1.5 ml Bioruptor Plus TPX microtubes in a Bioruptor Plus instrument (Diagenode, Denville, NJ, USA). The quality of RNAs and the size of fragmented RNAs were analyzed on a Bioanalyzer 2100 (Agilent, Santa Clara, CA, USA). 1 µg of total fragmented RNAs was stored at −80°C for RNA-seq. 4sU-labeled RNAs were purified from 600 µg of total fragmented RNA. Biotinylation and purification of 4sU-labeled RNAs were performed as described (Dölken et al, 2008; Gressel et al, 2019a). Separation of 4sU-labeled RNAs was carried out with streptavidin beads (Miltenyi Biotec, Bergisch Gladbach, Germany). Prior to library preparation, total fragmented RNAs and 4sU-labeled RNAs were treated with DNase (Qiagen, Hilden, Germany), column purified (miRNeasy Micro Kit, Qiagen, Hilden, Germany), and quantified using a Qubit Fluorometer (Invitrogen/Thermo Fisher Scientific, Waltham, MA, USA). The quality of RNAs was analyzed on a Bioanalyzer 2100 (Agilent, Santa Clara, CA, USA). Strand-specific libraries of total fragmented RNAs (RNA-seq) and 4sU-labeled RNAs were prepared with the Ovation Universal RNA-Seq System (NuGEN/Tecan, Männedorf, Switzerland) using random hexamer priming only according to the manufacturer's instruction with minor modifications (Gressel et al, 2019a). The size selected libraries were analyzed on a Bioanalyzer 2100 (Agilent, Santa Clara, CA, USA) before paired-end sequencing on the Illumina HiSeq 2500. TT-Seq libraries were sequenced to the depth of ∼ 100 million uniquely mapped reads.

## CRISPR/Cas9-mediated enhancer deletion in DU 145 cells

CRISPR/Cas9-mediated deletion of the MYC enhancer region on chromosome 8q24 (GRCh37/hg19, chr8: 128226403-128746490) and the Immunoglobulin Heavy (IGH) gene locus on chromosome 14q32.33 (GRCh37/hg19, chr14: 106527004-107035452) were carried out in DU 145 cells. The single guide RNAs (sgRNAs) were designed (http://www.broadinstitute.org/rnai/public/analysis-tools/sgrna-design) to span the entire MYC enhancer region and IGH locus, respectively (Eurofins MWG Operon). sgRNAs were cloned into an sgRNA Cloning Vector (Addgene Plasmid #41824, RRID:SCR_002037) using Gibson assembly master mix (NEBuilder HiFi DNA assembly Master Mix, Cat no. E2621S; NEB, Ipswich, MA, USA). DU 145 ($2 \times 10^6$) cells were transfected (using FuGENE HD Transfection Reagent, Cat.no E2312; Promega, Madison, WI, USA) with 10 µg of eight pooled equimolar sgRNA constructs together with hCas9 plasmid (Addgene Plasmid # 41815, RRID:SCR_002037). Post-transfection half of the cultured cells were collected for PCR genotyping, while the other half was re-plated for culturing. Cells were collected at days 2, 4, and subsequently every fourth day till day 32. DNA from cells was extracted (using DNeasy Blood & Tissue Kit, Cat. no. 69506; Qiagen, Hilden, Germany) and genotyped with 300 ng of DNA at following conditions—initial denaturation at 95°C for 5 min; denaturation at 98°C for 15 s, annealing at 60°C for 30 s, extension at 72°C for 30 s (31 cycles for MYC, 38 cycles for IGH, and 24 cycles for GAPDH genotyping); and final extension at 72°C, 5 min. Note, the combination of the four terminal sgRNAs can lead to different lengths of the MYC enhancer deletion region, which can be subsequently detected with genotyping primers (725-, 741-, 810-, or 826-bp PCR product). Each experiment was done in triplicate. Sequences of guide RNAs and primer pairs are given in Appendix Table S2. sgRNA location at the MYC enhancer locus is shown in Dave et al (2017).

## TT-seq and RNA-seq data preprocessing

Paired-end 50-bp reads were first mapped to a single copy of the rDNA locus to remove rRNA-related sequences. Reads that did not map to the rDNA were then aligned to the hg20/hg38 (GRCh38) genome assembly (Human Genome Reference Consortium) using STAR 2.6.0c (Dobin et al, 2013) with the following specifications: outFilterMismatchNoverLmax 0.05, outFilterMultimapScoreRange 0, and alignIntronMax 500,000. Bam files were filtered with SAMtools (Li et al, 2009) to remove alignments with MAPQ smaller than 7 (-q 7), and only proper pairs (-f99, -f147, -f83, -f163) were selected. Fragment counts for different features were calculated with HTSeq (Anders et al, 2015). Further data processing was carried out using the R/Bioconductor environment.

## Normalization of TT-seq samples

To normalize TT-seq samples for sequencing depth and make them comparable between cell lines, we used the "median of ratios" approach described in the DESeq2 R/Bioconductor package (Anders & Huber, 2010). Briefly, TT-seq fragment counts and coverage tracks were divided by sample-specific size factors determined by the median ratio of gene counts relative to the geometric mean per gene, estimated on GENCODE (v31) gene counts. Hereafter, we refer to such size factor normalized TT-seq fragment counts and coverage tracks simply as normalized counts and normalized coverage, respectively. For within-sample comparison, normalized transcript counts were further normalized by transcript length to derive normalized FPK (fragments per kilobase of transcript) values.

## TT-seq expression cutoff estimation

To determine a suitable cutoff to distinguish transcribed genomic regions from background, we plotted the densities of replicate-averaged $\log_2$(normalized FPK) values over all GENCODE (v31) annotated genes (type = gene). Distributions were generally bimodal, and a suitable cutoff was selected in the valley between the two peaks. Since distributions looked overall similar between samples, we selected normalized FPK $\geq 20$ as expression cutoff for all samples. Unless stated otherwise, this cutoff was used to distinguish between transcribed (active) and untranscribed (inactive) genomic regions.

## Transcription unit annotation and classification from TT-seq data

Transcription unit annotation and classification were done as described (Michel et al, 2017) with a few modifications. Briefly, strand-specific coverage was calculated from fragment midpoints in consecutive 200-bp bins throughout the genome for all TT-seq samples. Binning reduced the number of uncovered positions within expressed transcripts and increased the sensitivity for detection of lowly synthesized transcripts. The GenoSTAN R/Bioconductor package (Zacher et al, 2017) was used to learn a two-state hidden Markov model with a PoissonLog-Normal emission distribution in order to segment the genome into "transcribed" and "untranscribed" states, which resulted in 51,198–134,528 TUs per cell line. TUs that overlapped at least 25% of a protein-coding gene annotated in RefSeq (Release 109.20190607) or GENCODE (v31) and

overlapped with an annotated exon of the corresponding gene were classified as mRNAs. Remaining TUs were annotated as non-coding (nc)RNAs. In order to overcome low expression or mappability issues, ncRNAs that were only 200 bp (1 bin) apart were merged. Subsequently, TU start and end sites were refined to nucleotide precision by finding borders of abrupt coverage increase or decrease between two consecutive segments in the two 200-bp bins located around the initially assigned start and stop sites via fitting a piece-wise constant curve to the coverage profiles (whole fragments) for all TT-seq samples using the segmentation method from the R/Bioconductor package tilingArray (Huber *et al*, 2006). In order to filter spurious predictions, TUs were further filtered with a minimal expression threshold of replicate-averaged normalized FPK ≥ 20 as described above (requiring at least 15 normalized FPK per replicate). ncRNAs overlapping with GENCODE (v31) annotated small non-coding RNA classes (snRNA, snoRNA, tRNA) were omitted from further analysis. The remaining ncRNAs were further classified according to their genomic location relative to protein-coding genes into four categories: upstream antisense RNA (uaRNA), convergent RNA (conRNA), antisense RNA (asRNA), and intergenic RNA. ncRNAs located on the opposite strand of an mRNA were classified as asRNA if the TSS was located > 1 kbp downstream of the sense TSS, as uaRNA if the TSS was located < 1 kbp upstream of the sense TSS, and as conRNA if the TSS was located < 1 kbp downstream of the sense TSS. Each of the remaining ncRNAs was classified as intergenic.

eRNA annotation (see also Fig 1B): Of the ncRNA classes defined above, we further classified intergenic and asRNAs as eRNAs, if their TSS ± 500 bp overlapped with an enhancer state annotated by GenoSTAN (Zacher *et al*, 2017) based on patterns of five histone modifications (H3K4me1, H3K4me3, H3K36me3, H3K27me3, and H3K9me3) in any of 127 cell types and tissues covering 111 datasets of the Roadmap Epigenomics Project (Kundaje *et al*, 2015) as well as 16 datasets of the Encyclopedia of DNA Elements (ENCODE) Project (Dunham *et al*, 2012). The ± 500-bp window was chosen to be more permissive, due to the limited resolution of ChIP-seq and since for very lowly expressed transcripts the starts of TT-seq-defined TUs may deviate from the actual TSS. Long antisense eRNAs that overlapped with the promoter region of the respective protein-coding gene (TSS ± 1 kbp) were annotated to end 1 kbp downstream of the protein-coding TSS. Next, we introduced additional filtering criteria, as highly transcribed TUs can give rise to spurious downstream TUs, which do not represent independent TUs, but ongoing, gradually decreasing transcription by Pol II downstream of the main TU. To avoid misclassification of such downstream TUs as putative eRNAs, we did the following:

First, we restricted our set of intergenic eRNAs to only those originating from regions with TT-seq detected transcription on both strands, with an allowed gap of up to 750 bp between divergently transcribed eRNA pairs. To this end, a non-coding TU had to be annotated on the opposite strand, but it was not required to lie above the expression threshold. The permissive 750-bp gap width cutoff was based on 5′-GRO-seq data presented in Duttke *et al* (2015) (Fig 1C) where they report the distribution of distances between divergent TSSs to be ~ 200 bp on average and ~ 700–750-bp maximal.

Second, we also restricted our set of antisense eRNAs. To this end, non-coding TUs downstream of mRNA/uaRNA TUs were sequentially merged with the upstream mRNA/uaRNA if the gap between TUs was < 5 kbp. Subsequently, non-coding TUs, which had been classified as antisense eRNAs and were constituents of such merged regions, were excluded from further analyses, unless their TT-seq signal (normalized FPK) was at least 2-fold higher than TT-seq signal over the end (last 1 kbp) of the upstream mRNA/uaRNA.

Enhancer annotation (see also Fig 1C): Based on the eRNA annotation, we next approximated putative enhancer regions, which are the regions from which eRNA synthesis is initiated. To define intergenic enhancer regions, we selected the region between the TSSs of each pair of intergenic bidirectional eRNAs and extended it by 500 bp in both directions. To define intragenic enhancer regions, we selected the region covering 750 bp upstream to 500 bp downstream of the TSS of antisense eRNAs. Since we allowed a gap of maximum 750 bp between bidirectional intergenic eRNAs, we selected 750 bp also as putative enhancer region upstream of antisense eRNAs. Based on this definition, all intragenic enhancer regions were 1,250 bp long whereas the length of intergenic enhancer regions was dependent on the gap or overlap between bidirectional eRNAs. Strongly transcribed intergenic enhancer regions gave rise to uninterrupted bidirectional TT-seq signal over long genomic stretches, which could not be resolved at the resolution of individual enhancers, and did therefore result in the definition of some enhancer regions covering several kbp.

Dealing with annotated lncRNAs: The FANTOM consortium has shown that the majority of intergenic lncRNAs originate from enhancer regions (Hon *et al*, 2017), and thus, we only excluded eRNAs/enhancers if they overlapped a functional lncRNA originating from an intergenic promoter DHS (24 lncRNAs, e.g., Malat1; taken from Supplementary table 5 in Hon *et al* (2017) selecting CAT_geneCategory = p_lncRNA_intergenic).

## Comparison of TT-seq-defined eRNA/enhancer annotations with other genomic enhancer marks

To compare TT-seq-defined eRNA/enhancer annotations with other genomic enhancer marks, we made use of published data for K562, HCT 116, and LoVo cell lines (Figs 1 and EV1, Appendix Fig S1B–E).

### K562
To obtain read coverage, DNase I(DHS)-seq and all ChIP-seq data were taken from ENCODE via GEO (DHS = GSM736629, GSM736566 (Thurman *et al*, 2012); H3K4me1 = GSM733692; H3K27ac = GSM733656 (Dunham *et al*, 2012)), and PRO-seq/GRO-cap data were taken from Core *et al* (2014) via GEO (PRO-seq = GSM1480237; GRO-cap = GSM1480321). bigWig files were downloaded and lifted from hg19 to hg38 human genome build. In case of DHS, there was one bigWig file per biological replicate and we summed the signal over both replicates. In case of GRO-cap, we downloaded raw sequence reads, mapped them against the hg38 genome assembly using Bowtie (version 2.3.4.1) (Langmead *et al*, 2009), and considered only the first base for calculating the coverage.

### HCT 116
We downloaded hg38 bigWig files from the ENCODE portal (H3K4me1 = ENCFF774BWO; H3K27ac = ENCFF984WLE; DHS =

ENCFF169PCK (Zhang *et al*, 2020)) to show as signal tracks in UCSC genome browser (Fig 1B, Appendix Fig S1B).

### LoVo

To obtain read coverage, DNase I(DHS)-seq and H3K27ac ChIP-seq data were downloaded from GEO (DHS = GSM2400377, GSM2400378 (Dunham *et al*, 2012); H3K27ac = GSM3592807, (Hung *et al*, 2019)). Reads were aligned to the hg38 genome assembly using Bowtie (version 2.3.4.1) (Langmead *et al*, 2009). Coverage per bp was calculated for single-end ChIP-seq data after extending reads by 200 bp and for paired-end data using physical coverage, that is, counting both sequenced bases covered by reads and unsequenced bases spanned between proper mate-pair reads. In case of DHS, coverages for both replicates were summed. H3K27ac signal surrounding the TSS was calculated as mean $\log_2$(read coverage + 1) within ± 500 bp (Fig EV1B).

Transcription factor ChIP-seq peak regions were downloaded from CISTROME (Zheng *et al*, 2019) on July 16, 2018. For the LoVo cell line, this comprised peak regions for 326 distinct TFs (all published in Yan *et al* (2013)), for which we calculated the genome wide per bp coverage as the sum of distinct TF peak regions overlapping each genomic position (Figs 1F, 7 and EV4). Such TF peak region coverage was also calculated for HCT 116 cells, containing data for 19 distinct TFs (Fig 4E). Individual peak regions shown in Fig 4E are YY1 (CistromeDB 46204), TCF7l2 (CistromeDB: 45714), SP1 (CistromeDB: 46215), EP300 (CistromeDB: 42907), and POLR2A (CistromeDB: 4620).

### Quantification of eRNA synthesis and enhancer transcription over all cell lines

To compare eRNA synthesis and enhancer transcription across cell lines, we first generated merged eRNA and enhancer annotations containing all eRNAs and enhancers annotated in at least one of the cell lines. This was done using the reduce() function from the GenomicRanges R/Bioconductor package over concatenated eRNA and enhancer annotations from all cell lines. These merged annotations comprised 63,216 eRNAs and 58,457 enhancers (Table EV6). To quantify eRNA synthesis, we counted TT-seq fragments over the annotated eRNA transcripts (see above) and derived normalized FPK values. To quantify enhancer transcription, we counted TT-seq fragments over the annotated enhancer regions (see above), considering only counts on the antisense strand, in case of enhancers defined by antisense (i.e., intragenic) eRNAs, and summing the counts over both strands in case of enhancers defined by bidirectional intergenic eRNAs. To derive normalized FPK values, TT-seq fragment counts were further normalized by enhancer length in case of intergenic enhancers and by enhancer length −750 bp in case of intragenic enhancers.

### Quantification of mRNA synthesis over all cell lines

To compare mRNA synthesis across cell lines, we calculated normalized FPK values for all RefSeq protein-coding genes (NM_, Release 109.20190607). 14,627 RefSeq protein-coding genes were transcribed in at least one cell line.

### Transcription directionality analysis

To investigate transcription directionality at promoters, we took all protein-coding genes annotated in RefSeq (NM_, Release 109.20190607) and selected only those for which all cell lines used the same promoter region and no protein-coding gene was annotated on the antisense strand within ± 5 kbp of the sense TSS. In each cell line, for each of these promoters, we calculated normalized TT-seq fragment counts in 1 kbp regions downstream (sense, TSS to TSS + 1,000 bp) and upstream (antisense, TSS − 1,150 bp to TSS − 150 bp) of the protein-coding TSS. Promoters were classified as bidirectional or unidirectional if at least 10 counts were observed on both strands or the protein-coding strand only, respectively. For all bidirectional promoters, we defined a transcription directionality score as the ratio of mRNA and uaRNA counts. To compare protein-coding gene transcription at unidirectional vs bidirectional promoters, we compared normalized TT-seq fragment counts in 1 kb regions downstream of the protein-coding TSS. To compare protein-coding gene transcription at promoters which were unidirectional in all cell lines vs promoters which were bidirectional in at least one of the cell lines, we compared the maximum normalized TT-seq fragment counts over all cell lines. To investigate whether directionality scores were preserved across cell lines, we correlated uaRNA and mRNA counts at bidirectional promoters over cell lines. Only promoters with observed bidirectional transcription in at least 5 cell lines and varying transcription over cell lines (coefficient of variation > 50%) were considered ($n = 3,468$), and Pearson correlation coefficients $r$ were calculated between vectors of TT-seq $\log_2$(counts) representing uaRNA and mRNA synthesis patterns across the cell lines.

To investigate transcription directionality at enhancers, we took all bidirectional intergenic enhancer regions shorter than 3 kbp from the unified enhancer annotation (see above) and calculated normalized TT-seq fragment counts in these regions on the plus and minus strand (relative to the plus strand of the genomic reference sequence), only considering cell lines in which the respective enhancers were annotated. Based on this, we defined a transcription directionality score as the ratio of plus and minus strand counts. To investigate whether directionality scores were preserved across cell lines, we correlated plus and minus strand transcription at bidirectional enhancers over cell lines. Only enhancers with observed bidirectional transcription in at least 5 cell lines and varying transcription over cell lines (coefficient of variation > 50%) were considered ($n = 715$), and Pearson correlation coefficients $r$ were calculated between vectors of TT-seq $\log_2$(counts) representing plus and minus strand transcription patterns across the cell lines.

### Comparison to putative enhancers called by other methods

We downloaded FANTOM5 (CAGE) transcribed enhancers (Andersson *et al*, 2014) in hg38 (F5.hg38.enhancers.expression.usage.matrix) from http://doi.org/10.5281/zenodo.556775 (Rennie *et al*, 2018). CAGE enhancer calls were available for four of the cell lines investigated here: K562, Jurkat (CNhs11250), DU 145 (CNhs11253), and PC-3 (CNhs11260). K562 CAGE enhancer calls were available for three biological replicates (CNhs12334, CNhs12335, CNhs12336), and overlap calculations were done for both the union and the intersection of enhancer calls from the three

replicates (Appendix Fig S1F). Jurkat ChRO-seq bigWig files for two biological replicates were downloaded from GEO (GSM3309956, GSM3309957, (Chu *et al*, 2018)), and transcription initiation regions (TIRs) were called using the dREG gateway (Wang *et al*, 2019). Resulting TIRs were lifted from hg19 to hg38 human genome build, and only regions annotated in both replicates were considered further.

To allow for a better comparison of CAGE and ChRO-seq candidate enhancers with TT-seq candidate enhancers, we further filtered CAGE enhancers and ChRO-seq TIRs using the same strategy as we used for filtering TT-seq enhancers: We selected only regions with at least 1 kbp distance from protein-coding TSSs, not overlapping annotated small non-coding RNA classes (snRNA, snoRNA, tRNA), not overlapping a functional lncRNA originating from an intergenic promoter DHS, and overlapping an enhancer state annotated by GenoSTAN (see TT-seq eRNA/enhancer definition for details).

## Comparison to high-throughput functional assays

For HCT 116 and K562 cells, we downloaded STARR-seq peak .bed files from the ENCODE portal (HCT 116: ENCFF305HQC, ENCFF594EEA; K562: ENCFF717VJK, ENCFF394DBM (preprint: Lee *et al*, 2020)). For each of the two cell lines, we considered only peak regions called in both of the two biological replicates (HCT 116: $n = 28{,}771$; K562: $n = 20{,}552$). To also compare STARR-seq overlap with candidate enhancers defined by other methods, we took HCT 116 and K562 DNase I(DHS), H3K4me1, and H3K27ac peak regions from ENCODE (HCT 116: DHS = GSM736600, GSM736493 (Thurman *et al*, 2012), H3K4me1 = ENCFF986BGX, H3K27ac = ENCFF349LKU (Zhang *et al*, 2020); K562: DHS = GSM736629, GSM736566 (Thurman *et al*, 2012), H3K4me1 = GSM733692, H3K27ac = GSM733656 (Dunham *et al*, 2012); for ChIP-seq, we downloaded replicatedPeak regions and for DHS took the intersection of broadPeak regions from both replicates and filtered them using the same strategy as we used for filtering TT-seq enhancers (see *Comparison to putative enhancers called by other methods*). To calculate enrichments over expected STARR-seq overlap, we randomly sampled 20 sets of TT-seq-matched enhancer regions. To this end, random non-coding regions (same number as observed TT-seq enhancers) were chosen to have a similar overall distribution of their widths and distances from nearest gene promoters, which had a similar overall distribution of transcription levels and cell type specificity scores. Such null sets were generated separately for TT-seq-defined enhancers in HCT 116 and K562, and STARR-seq overlaps determined to obtain empirical null distributions and to calculate empirical P-values. To investigate whether candidate enhancers not called active in STARR-seq were enriched for having annotated neighboring enhancers in close proximity, we counted the number of neighboring enhancers within $\pm$ 12.5 kbp for each candidate enhancer of a conservative ("combined") set of enhancers annotated by DHS, H3K27ac and TT-seq. The 12.5 kbp cutoff was chosen as it is also the distance used for stitching together enhancers before super-enhancer calling (see below) and as we observed that 85% of Med1 called super-enhancer regions in LoVo (Fig 4A) have a length shorter than 25 kb.

For K562, we downloaded annotations for CRISPRi tested and functionally verified enhancers from two studies: In Schraivogel *et al* (2020), they targeted 1,786 ENCODE-predicted enhancers, of which 77 had a significant effect on neighboring gene expression (hg19 enhancer coordinates were taken from their published tables S2 and S3). In Gasperini *et al* (2019), they targeted 5,779 enhancers, of which 600 had a significant effect on neighboring gene expression (hg19 enhancer coordinates were taken from their published tables S2A and B). Annotations were lifted from hg19 to hg38 human genome build before calculating overlaps with TT-seq or STARR-seq enhancer regions.

## Super-enhancer analysis

To identify super-enhancers (SEs) in LoVo cells, we downloaded published Med1 ChIP-seq data (Yan *et al*, 2013), mapped it to the hg20/hg38 (GRCh38) genome assembly using Bowtie (version 2.3.4.1) (Langmead *et al*, 2009) and defined SEs as described (Whyte *et al*, 2013). For the comparison of transcriptional activity at typical enhancers (TEs, i.e., Med1 peaks not classified as SE) and SEs, we obtained FPK values for the respective regions. In case a Med1 enhancer region was stitched together from individual Med1 peaks separated by less than 12.5 kbp (Whyte *et al*, 2013), we calculated FPK values by considering only the constituent peak regions. For comparison of eRNA lengths, we took all eRNAs annotated in LoVo and divided them into two groups: those originating from a SE region (i.e., eRNA TSS overlapping a SE) and those originating outside of SE regions.

## Cell type specificity of mRNA and eRNA synthesis

We defined two types of entropy-based cell type specificity scores.

1   Overall cell type specificity ranks a promoter/enhancer according to the degree to which its transcriptional activity differs from ubiquitous (i.e., transcribed at any level above background in all cell lines) uniform expression. We calculated overall cell type specificity as described (Schug *et al*, 2005; Andersson *et al*, 2014; Hirabayashi *et al*, 2019):

$$\mathrm{Specificity\,score}(p) = 1 - \frac{H(p)}{\log_2(n)}$$

where $H$ is the entropy of a discrete probability distribution:

$$H(p) = -\sum_{i=1}^{n} p_i \log(p_i); p_i = \frac{x_i}{\sum_{i=1}^{n} x_i}$$

where $x_i$ is the transcriptional activity (normalized FPK) of a promoter or enhancer in cell line $i$, and $n$ is the number of cell lines. The specificity score ranged between 0 and 1, where 0 means unspecific (i.e., uniform synthesis across cell lines) and 1 means cell type-specific synthesis.

2   Specificity of a promoter/enhancer's transcription pattern with respect to a particular cell line ranks a promoter/enhancer according to the similarity its transcription pattern has toward synthesis in only that particular cell line. This cell type specificity score was calculated as described (Cabili *et al*, 2011) and relies on Jensen–Shannon (JS) divergence, with a perfect cell type-specific pattern having JS = 1. The JS divergence of two discrete probability distributions, $p^1$, $p^2$, is defined to be

$$\mathrm{JS}\left(p^1, p^2\right) = H\left(\frac{p^1 + p^2}{2}\right) - \frac{H(p^1) + H(p^2)}{2}.$$

Relying on the theorem that the square root of the JS divergence is a metric (Fuglede & Topsoe, 2004), we define the distance between two across-cell-type transcription patterns, $t^1$ and $t^2$, and $t^i = (t_1^i, \cdots, t_n^i)$, as

$$\mathrm{JS}_{\mathrm{dist}}\left(t^1, t^2\right) = \sqrt{\mathrm{JS}(t^1, t^2)}.$$

The cell line specificity of a promoter/enhancer's transcription pattern, $t$, across $n$ cell lines with respect to cell line $c$ can then be defined as

$$\mathrm{JS}_{\mathrm{sp}}(t|c) = 1 - \mathrm{JS}_{\mathrm{dist}}(t, t^c)$$

where $t^c$ is a predefined transcription pattern that represents the extreme case in which a promoter/enhancer is transcribed only in cell line $c$.

$$t^c = \left(t_1^c, \cdots, t_n^c\right), \text{ such that } t_i^c = \begin{cases} 1 & \text{if } i = c \\ 0 & \text{otherwise} \end{cases}.$$

Finally, the overall specificity score was used for Fig 5C and for generating matched sets of randomly sampled E–P pairs (Fig 6F). For all other analyses and figures, the specificity score with respect to a particular cell line was used.

### Enhancer–promoter pairing

To obtain putative E–P pairs, we used pairwise across-cell-type correlations between enhancer and promoter transcriptional activity. With the "correlated neighboring" (CN) and "correlated window" (CW) approach, enhancers were paired to the nearest upstream and downstream neighboring active promoters, and all active promoters within ± 500 kbp distance, respectively. For E–P pairing, active promoters were defined by having normalized FPK ≥ 30. E–P distances were calculated between enhancer midpoints and gene TSSs. Only pairs with correlated transcriptional activity ($r > 0.6$) were kept. For this, Pearson correlation coefficients r were calculated between vectors of TT-seq $\log_2$(normalized FPK + 1) values representing transcriptional activity patterns across the 14 cell lines.

As a third approach, we also paired enhancers to their nearest active gene, "nearest" (N). Only E–P pairs spanning a distance of less than 500 kbp were considered. This approach was used in addition to also provide putative E–P pairing information in the absence of correlated transcriptional activity. Moreover, this approach was used to show that cell type-specific enhancers are associated with cell type-specific gene expression (Fig 5D–F), as such analysis would have been confounded by circular reasoning if correlated E–P pairs would have been used.

To show that cell type-specific enhancers were associated with cell type-specific gene activity, we paired enhancers either to their nearest active promoter (Fig 5D and E) or to interacting active promoters (Appendix Fig S4D and E). For the latter, we used promoter capture Hi-C data (LoVo cells, see next paragraph) or Pol II ChIA-PET data (K562 cells). We combined published K562 Pol II ChIA-PET interactions from ENCODE (ENCSR000BZY (Krismer

et al, 2020)) and (Li et al, 2012). Respective .bed files were downloaded from GEO (GSM970213, GSM832463, GSM832464, GSM832465) and lifted from hg19 to hg38 human genome build.

### Comparison of enhancer–promoter pairs with CHi-C data

Published promoter capture Hi-C data for the LoVo colorectal cancer cell line (Orlando et al, 2018) were obtained from the European Genome-phenome Archive and preprocessed as described in the publication, but using hg20/hg38 (GRCh38) as reference genome. Interactions with a score ≥ 3.0 (obtained by CHiCAGO (Cairns et al, 2016)) were considered for further analyses. We defined that an E–P pair was supported by a CHi-C interaction if we observed a CHi-C interaction for which the bait fragment overlapped the promoter and the hit fragment overlapped the enhancer of a particular E–P pair. Only E–P pairs with distance greater than 10 kbp were used to investigate support by CHi-C interactions because shorter range interactions cannot reliably be identified in the CHi-C data. To calculate enrichments over expected values of supported interactions, we randomly sampled 100 sets of matched E–P pairs. To this end, we first drew promoters (same number as observed E–P pairs) with a similar overall distribution of transcription levels and cell type specificity scores and then paired them to random non-coding regions of the genome with similar overall distribution of their widths and distances from gene promoters. Such null sets were generated separately for E–P pairs obtained with each of the three pairing approaches and CHi-C interaction overlaps determined to obtain empirical null distributions and to calculate empirical P-values.

### Comparison of enhancer–promoter pairs with pairs predicted by the activity-by-contact model

Published predictions for E–P interactions in K562 cells Fulco et al, 2019) were downloaded from https://osf.io/uhnb4/. Putative enhancer annotations from K562.AllPredictions.txt and K562.PositivePredictions.txt were lifted from hg19 to hg38 genome build and then overlapped with K562 TT-seq enhancer annotations. 68% of TT-seq candidate enhancer regions were also tested by the ABC model. Since one of our pairing approaches pairs enhancers to the nearest active gene promoter (within 500 kbp), almost all of the putative enhancers reported in our study pair to at least one promoter. This resulted in a considerable fraction (59%) of putative enhancers which were paired in our study but not paired by the ABC model. Thus, we took only positive E–P pairs reported by the ABC model and compared them to E–P pairs described in our study for the same enhancers.

### Enrichment analysis of gene classes in E–P pairs

Enrichment analysis was done for putative E–P pairs defined by the CN pairing approach. The following gene lists were considered for enrichment analysis: housekeeping genes ($n = 3{,}804$ (Eisenberg & Levanon, 2013)), transcription factors ($n = 1{,}672$ (Abugessaisa et al, 2016)), and cancer-associated genes ($n = 723$, Cancer Gene Census_V91 (Sondka et al, 2018)). We used Fisher's exact test on $2 \times 2$ contingency tables to compute P-values.

## GWAS SNP enrichment analysis

All SNP-disease/trait associations were taken from the NHGRI-EBI GWAS Catalog Welter *et al*, 2014) (http://www.ebi.ac.uk/gwas/) downloaded on April 18, 2020. SNPs were filtered to retain only those located in the non-coding genome (i.e., intronic and intergenic) ($n = 100,108$). We further obtained a list of SNPs associated with cancer (EFO_0000311) by GWAS and from this selected all traits corresponding to the types of cancer cell lines used in this studys (Table EV7, $n = 2,312$, hereafter referred to as cancer-associated SNPs). For comparison of the overlap of cancer-associated SNPs and enhancers to the overlap of SNPs from the 1000 Genomes project (Phase 3v5) and enhancers, we randomly sampled 1,000 matched SNP sets from the EUR population (with the same size as the original cancer-associated SNP set), using SNPsnap (Pers *et al*, 2014) with default parameters (except $r^2 = 0.8$). SNPsnap matched the original SNPs based on MAF, gene density, distance to nearest genes, and number of sites within LD of 0.8 of the original site. SNPs were then overlaid with TT-seq-defined enhancer regions, and the P-value was determined based on the empirical background distribution.

## Data availability

The raw and processed TT-seq data together with the eRNA and enhancer annotations have been deposited in the Gene Expression Omnibus (GEO) database under accession code GSE152291 (https://www.ncbi.nlm.nih.gov/geo/query/acc.cgi?acc=GSE152291).

**Expanded View** for this article is available online.

## Acknowledgements
We thank Helmut Blum and Stefan Krebs (LAFUGA, Gene Center Munich) and Kerstin Maier (Max Planck Institute for Biophysical Chemistry) for sequencing. We are deeply grateful to Björn Schwalb, Johannes Söding, Inderpreet Kaur Sur, Åsa Kolterud, Carina Demel, Marieke Oudelaar, and Alan Cheung for helpful advice and discussion. We would like to thank Björn Schwalb and Kristina Žumer for critical reading of the manuscript. We thank Vladyslav Dembrovskyi and Simon Wengert for help with initial data analyses. We acknowledge the Capture Hi-C data set provided by The Institute of Cancer Research (ICR)/ Molecular and Population Genetics at ICR. KL, LAJ, and ML were funded by the Center for Innovative Medicine (CIMED) at Karolinska Institutet and by the Science for Life Laboratory (SciLifeLab) in Stockholm. PC was supported by the Deutsche Forschungsgemeinschaft within SFB860 and SPP1935 and under Germany's Excellence Strategy (EXC 2067/1-390729940), the European Research Council Advanced Investigator Grant TRANSREGULON (grant agreement No 693023), CIMED, SciLifeLab, and the Volkswagen Foundation. Open Access funding enabled and organized by ProjektDEAL.

## Author contributions
KL designed and carried out TT-seq experiments, contributed to, and interpreted bioinformatics data analysis. LAJ carried out TT-seq experiments and contributed to data interpretation. HvdE processed ChIP-seq and CHi-C data and conducted Med1 super-enhancer analysis. KD carried out CRISPR/Cas9-mediated enhancer deletion. JT provided conceptual input, and contributed data and material. ML designed, conducted, and interpreted bioinformatics analysis. ML and PC supervised research. KL, ML, and PC wrote the manuscript, with input from all authors.

## Conflict of interest
The authors declare that they have no conflict of interest.

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
