## [Review Process File · Molecular Systems Biology]

Transcriptionally active enhancers in human cancer cells

Katja Lidschreiber, Lisa Jung, Henrik von der Emde, Kashyap Dave, Jussi Taipale, Patrick Cramer, and Michael Lidschreiber

DOI: 10.15252/msb.20209873

Corresponding author(s): Michael Lidschreiber (michael.lidschreiber@mpibpc.mpg.de) , Patrick Cramer (patrick.cramer@mpibpc.mpg.de)

Review Timeline:

Submission Date:	17th Jul 20
Editorial Decision:	13th Aug 20
Revision Received:	11th Nov 20
Editorial Decision:	4th Dec 20
Revision Received:	11th Dec 20
Accepted:	16th Dec 20

Editor: Jingyi Hou

Transaction Report:

Thank you for submitting your work to Molecular Systems Biology. We have now heard back from the three reviewers who agreed to evaluate your manuscript. As you will see below, the reviewers acknowledge the potential interest of the study. They raise however a series of concerns, which we would ask you to address in a major revision.

I think that the reviewers' recommendations are rather clear and there is no need to reiterate their comments. In particular, some of the identified enhancers need to be experimentally validated as suggested by reviewer #1. All other issues need to be addressed as well. As you may already know, our editorial policy allows in principle a single round of major revision, so it is essential to provide responses to the reviewers' comments that are as complete as possible. Please feel free to contact me in case you would like to discuss in further detail any of the issues raised by the reviewers.

On a more editorial level, please do the following.

REFEREE REPORTS

Reviewer #1:

In this manuscript Lidschreiber et al. have applied TT-seq to annotate transcribed enhancers in 14 cancer lines, and utilized this information to identify enhancer characteristics and relation to cell type specific promoter activity. They have quantified enhancer and promoter transcription from TT-seq data and have done a commendable job in annotating eRNAs and enhancer regions. It is interesting to see how TT-seq could identify cell type specific enhancer transcription that were associated with cell type specific gene expression patterns. Their enhancer calls were coherent with known enhancer marks and also included functionally verified enhancers. In addition, they have utilized their transcription data from multiple cell lines to develop a 3-way approach that could computationally identify putative enhancer-promoter pairs by transcriptional correlation. This is a

useful strategy to identify target genes of enhancers and could benefit future studies. Finally, the authors have expanded the enhancer characterization to include their relation to SNPs in different cancer lines and found that cancer-associated SNPs were enriched in transcribed enhancers targeting cancer genes. Overall this work provides a refined analysis of enhancers across multiple cell types that could be beneficial for future functional studies, and also serves as a resource of enhancers with mutations and E-P pairing information of cancer genes. The manuscript is very well written and experiments properly executed. However, here are some comments that could improve the manuscript further.

1.The authors have explained clearly in Figure 1B (and methods) the definitions for different classes of RNAs used in this work. It will be good to see the relevant histone modifications for the genomic location shown in the figure. Also, how enriched are these marks at the TSS of the asRNA relative to the gene-body of the mRNA so that these can be confidently annotated as eRNAs and not just spurious transcription from the mRNA coding gene? Have they looked into other available PRO-seq/GRO-cap data to see if these TSSs have paused RNA polymerase? Are these asRNAs even capped?

2.Minor point: The enhancer states were defined based on five histone modifications. What was the reason to not include H3K27ac among them to annotate eRNA?

3.The authors should validate some of these putative enhancers (defined by asRNAs and intergenic RNAs expressing regions) from multiple cell lines for their activities using conventional reporter (luciferase) assays and compare with previously annotated enhancers for a relevant cell line.

4.It would be helpful for the reader if the authors explain their rationale for selecting TSS +/- 500 bp for histone modification overlap, to call eRNA. As we know, usually polymerase pauses by 60 bp from the TSS adjacent to the +1 nucleosome and I wonder if the authors have tried various size windows before deciding on 500 bp?

5.Minor point: It will be also nice to explain the rationale for annotating -500 to +750 of the asRNA as the enhancer region. Also, the space between the divergent TSS is ~300 bp (Tippens et al. Biorxiv 2019, Core et al. Nat. Genet. 2014), so I wonder why did the authors choose a limit of 750 bp as the gap between divergent TSS of intergenic eRNAs.

6.In Figure 3B, the authors show variable enhancer transcription across different cell types. However, have they verified if there is any genomic translocation occurring at this locus, resulting in loss of the enhancer region in any of these lines?

7.I am confused with Figure 3C. Is this browser shot from WT or deleted cells? Also, for Figure 3D what are the 3 bands for the top panel? At 32 days the gene for the MYC deletion appears to get depleted compared to GAPDH. Please explain.

8.In line 296 the authors claim that TT-seq based enhancer transcription can be a good proxy for enhancer activity. Figure 3 and the text suggest that the enhancer containing genomic region is necessary for the expression of MYC in GP5d cells and its deletion does not cause depletion of DU145 cells where the enhancer is untranscribed. However, this particular region might not be necessary for DU145 cell survival irrespective of its transcriptional status, and MYC expression could be dependent on other enhancers as well. This does not address if enhancer transcription is a good read-out of enhancer activity or not. A more direct way to address this is by doing a CRISPRi of this enhancer region to repress its transcription (for example, in GP5d cells) and then measure the MYC expression/cell proliferation. This will indicate if enhancer transcription itself is necessary for MYC expression and cell proliferation.

9.For Figure 4E, is the Med1 ChIP-seq data not available for HCT116 and therefore the authors had to show the H3K27ac as super-enhancer mark? How well could this distinguish super-enhancers from enhancers?

10.It is interesting to see how cell-specific enhancers drive cell-specific gene expression. With respect to Figure 5D, there have been examples where an enhancer does not activate the nearest

promoter rather acts on one at a particular distance. In addition to the current analysis, I wonder if the authors could use the high contact frequency target promoters for the exclusive enhancers (based on Hi-C/capture Hi-C, for available cell lines) instead of the nearest ones and measure target promoter transcription in the those cell lines.

11. The authors have provided a nice multi-way strategy to identify E-P pairs, that have been expanded to identify the pairing information of cancer-associated genes. As they have mentioned that many such programs exist to predict target genes for enhancers, it will be interesting to see how their system compares with methods like the ABC model for relevant cell-types (Fulco et al. Nat. Genet 2019) that uses enhancer activity (from DHS and H3K27ac peaks) and Hi-C contacts to predict target genes in a cell-type specific manner.

12. Overall, TT-seq based enhancer annotation and characterization is a significant addition to other genome-wide methods for mapping enhancers, especially in so many different cancers. However, it will be interesting to see a comparative analysis or overlap between enhancers called by TT-seq vs GRO-cap/ChRO-seq or CAGE in relevant cell lines.

Reviewer #2:

This manuscript reports a set of TT-seq data obtained from a collection of cancer cell lines. The emphasis is on genome-wide identification of putative enhancers from these data. The data appear to be of high quality, and the analyses are generally carefully done and clearly described. I found the manuscript easy to read and the figures are very clear. The logic that is followed is not very much out-of-the-box, and the insights obtained are a bit incremental. Nevertheless, there are some interesting observations and the data will be a very useful resource for the community.

Major points:

1. The elephant in the room is: does TT-seq really do a good job identifying functional enhancers? This is not a new problem: one can ask the same for most enhancer predictions based on histone marks, ATAC-seq, Hi-C methods, and MPRAs. Very few of these data have been properly benchmarked against a real gold standard, which is to delete a representative set of the predicted enhancers by CRISPR and measure the activity of the surrounding promoters. The current manuscript does not go further than anecdotal (and perhaps somewhat cherry-picked) support provided on pages 8-9.

I think it would go too far to demand a systematic CRISPR deletion test for the current manuscript, because it could easily take a year of work to do this for a sufficient number of predicted enhancers. But it would be good if the authors could emphasize this limitation, and resist the temptation to over-interpret their results, for example in title, abstract, bottom page 6, and bottom page 9, line 476, and the Discussion. It may be good to systematically add the word "putative" or "candidate" in front of the word "enhancer".

Perhaps as a partial benchmarking, the authors could overlay their K562 data with the CRISPR screen that Schraivogel et al, Nat Meth 2020 recently reported? They found only 4% of ENCODE-predicted "enhancers" to control a nearby gene, and it will be interesting to know whether TT-seq has added value in predicting functional enhancers.

2. I wonder what the impact is of the decision of the authors to discard all TT-seq signals that

overlap with protein-coding genes. I understand that this may be done because of overlapping gene transcription, but surely there are a lot of enhancers in introns that are missed this way? A rough estimate of the "blind" proportion of the genome would be helpful.

3. Likewise, the E-P pairing by correlation of the TT-seq signals lacks "hard" experimental validation. It is quite possible that many of these correlations are only correlations and not causal E->P effects. Again, this would require a lot of CRISPR deletions of predicted enhancers to check the impact on the predicted partner promoter. If it is the choice of the authors not to do such validations, then of course the value of this analysis remains to be seen. This caveat should then be emphasized clearly in the text, because certainly not everyone in the cancer field may be aware of it.

4. There seems to be some circular reasoning in the manuscript:

a. l181-182: "The region surrounding the TSS of eRNAs was enriched for the active enhancer mark H3K27ac" -- yes, but l169-174 describe that eRNAs were required to originate from sites defined by a set of histone marks, which surely are not independent from H3K27ac.

b. l183: "and for transcription factor binding as monitored by ChIP-seq" -- same issue, ChIP-seq peaks of TFs are strongly correlated with histone marks.

c. eRNAs were required to be bi-directional (l75-l176). Surely this causes a bias in the directionality scores (l239-...)?

Bas van Steensel - review assignment 27 July 2020; completed 10 Aug 2020, apologies for the delay! [It is my standard policy to sign and date ALL of my reviews, regardless of my opinions and recommendations. PLEASE DO NOT REMOVE THIS NOTE.]

Reviewer #3:

The paper by Lidschreiber et. al. analyzed TT-seq data in 14 cancer cell lines. The data all appears to be excellent quality, and was sequenced to a high read depth. Most of the data provided is new data. The data analysis was conducted in an expert manner, providing an atlas of transcription units, enhancers, and enhancer targets that will be used elsewhere. As a result, this manuscript describes an excellent resource that is very clearly written and will be an important contribution to the literature. I am in favor of publication, however I have a few comments that should be addressed:

* All of the cell lines studied were model cell lines, and some have been passaged for a very long time in tissue culture. The authors state that transcriptional signatures are in good agreement with primary tumors. While this is no doubt true to some extent, it's clear that there are a lot of differences between primary tumors and cell lines as well due to differences in microenvironment and genetic factors. This does not diminish my enthusiasm for the present study, but I would like the authors to discuss the disadvantages (and advantages!) of these cell models in more detail.

* The manuscript identified only ~3000-3500 enhancers in each cell line. This seems somewhat low based on reports from other assays. Is this a matter of sensitivity? Can the authors describe factors that could influence sensitivity? Related to this, TT-seq does not get any signal from a paused

RNA polymerase. Could this decrease sensitivity?

* The authors use a cutoff directionality score >3 as imbalance and preferentially transcribed on both strands. However, this directionality score appears to be defined in log-10 units. It seems as though a lot of the features with a score <3 may also be quite imbalanced, depending on one's perspective. Could the authors please clarify, or update their description.

Responses are in italics

We thank all three reviewers for their encouraging and very constructive suggestions, which have greatly improved the manuscript. We provide a point-by-point response to each of the reviewers' comments below.

Overall changes to the manuscript. We have:

- added support that TT-seq can reliably detect intragenic enhancers based on observed transcription antisense of annotated protein-coding genes (new Appendix Fig 1B-E)*
- added comparative analyses to candidate enhancers annotated by other methods and to validated enhancers based on published high-throughput functional assays (STARR-seq and CRISPRi screens) to corroborate the putative enhancers defined by TT-seq in our study (new Fig EV3, Appendix Fig 1F)*
- added three additional paragraphs to the discussion, addressing 1) the comparative analysis to published high-throughput functional assays, 2) the implications and limitations of the used enhancer-promoter pairing approaches, and 3) the disadvantages/advantages of using model cell lines*
- clearly emphasized in the abstract, in concluding sentences of each paragraph and throughout the discussion that all enhancers and enhancer-promoter pairs presented here are putative*
- addressed all further specific questions of the reviewers (as specified below), and rephrased text as suggested*

Reviewer #1:

In this manuscript Lidschreiber et al. have applied TT-seq to annotate transcribed enhancers in 14 cancer lines, and utilized this information to identify enhancer characteristics and relation to cell type specific promoter activity. They have quantified enhancer and promoter transcription from TT-seq data and have done a commendable job in annotating eRNAs and enhancer regions. It is interesting to see how TT-seq could identify cell type specific enhancer transcription that were associated with cell type specific gene expression patterns. Their enhancer calls were coherent with known enhancer marks and also included functionally verified enhancers. In addition, they have utilized their transcription data from multiple cell lines to develop a 3-way approach that could computationally identify putative enhancer-promoter pairs by transcriptional correlation. This is a useful strategy to identify target genes of enhancers and could benefit future studies. Finally, the authors have expanded the enhancer characterization to include their relation to SNPs in different cancer lines and found that cancer-associated SNPs were enriched in transcribed enhancers targeting cancer genes. Overall this work provides a refined analysis of enhancers across multiple cell types that could be beneficial for future functional studies, and also serves as a resource of enhancers with mutations and E-P pairing information of cancer genes. The manuscript is very well written and experiments properly executed. However, here are some comments that could improve the manuscript further.

We thank the reviewer for the encouraging summary of our work, and for the detailed and constructive suggestions below.

1. The authors have explained clearly in Figure 1B (and methods) the definitions for different

classes of RNAs used in this work. It will be good to see the relevant histone modifications for the genomic location shown in the figure.

We thank the reviewer for this good suggestion. We agree that showing relevant histone modifications for the genomic location shown in Fig 1B will help the readers to relate TT-seq measured transcription with other marks that are often used to annotate enhancers. We have now added three additional tracks to Fig 1B: H3K27ac, H3K4me1, and DNase I signal (all taken from ENCODE). The signal of all three tracks agrees well with the TT-seq signal. Please note that the example originally shown in Fig 1B was from DU 145 cells, but since for this cell line not all of the additional data tracks were available we now show data for HCT 116 cells instead.

Also, how enriched are these marks at the TSS of the asRNA relative to the gene-body of the mRNA so that these can be confidently annotated as eRNAs and not just spurious transcription from the mRNA coding gene? Have they looked into other available PRO-seq/GRO-cap data to see if these TSSs have paused RNA polymerase? Are these asRNAs even capped?

These are valid concerns as with TT-seq we can map intragenic enhancer transcription only based on the TT-seq signal from the antisense strand. In Fig 1 (previously H&I, now E&F (right panels)) we already showed that the region upstream of antisense eRNA TSSs is enriched for DNase I and transcription factor ChIP-seq signal, indicating that antisense eRNAs are not just spurious transcription from the mRNA coding gene. To address the reviewer's concern further, we have now added four additional panels to Appendix Fig S1. First: in S1B we show 3 exemplary genomic loci of antisense eRNA transcription together with supportive signal tracks for relevant histone modifications (H3K27ac, H3K4me1) and DNase I signal.

Second: in S1C we compare the fraction of eRNAs overlapping with transcription factor ChIP-seq peak calls around their TSS for intergenic eRNAs vs antisense eRNAs in LoVo cells. The fractions were similar for both types of eRNAs.

Third: in S1D we show a metagene profile of PRO-seq signal aligned at and scaled between the TSS and end of antisense eRNAs. An enrichment of signal can be seen downstream of the TSS, however, it looks clearly less pronounced than the typical Pol II pausing peak usually observed at protein-coding genes. We also show a metagene profile of GRO-cap signal around the TSS of antisense eRNAs. The average GRO-cap signal peak aligned well with the TT-seq annotated TSSs.

Fourth: in S1E we show metagene profiles of H3K27ac, H3K4me1, and DNase I signal around the TSS of antisense eRNAs.

Together these results show that important marks to annotate enhancers are clearly enriched around the TSS of antisense eRNAs relative to the gene-body of the mRNAs, allowing us to confidently annotate them as eRNAs.

We have added the following reference to Appendix Fig S1 to the main text (page 6, line 183):

"... our putative eRNAs (hereafter also referred to as 'eRNAs') clearly showed properties characteristic of actively transcribed enhancers (Fig EV1B-D, Appendix Fig S1B-E)."

2.Minor point: The enhancer states were defined based on five histone modifications. What was the reason to not include H3K27ac among them to annotate eRNA?

Only five modifications (H3K4me1, H3K4me3, H3K36me3, H3K27me3, and H3K9me3) were available for all 127 epigenomes (Dunham et al., 2012; Kundaje et al., 2015)), which were then used by GenoSTAN to predict enhancer states for all 127 cell types. Therefore, the enhancer states used by us for annotating putative eRNAs were defined without including H3K27ac. However, we show in Fig 1 (previously Fig 1E, now Fig EV1B) that H3K27ac signal is clearly enriched at the TSSs of TT-seq defined putative eRNAs.

3. The authors should validate some of these putative enhancers (defined by asRNAs and intergenic RNAs expressing regions) from multiple cell lines for their activities using conventional reporter (luciferase) assays and compare with previously annotated enhancers for a relevant cell line.

We agree with the reviewer that investigating the functionality of some of our TT-seq predicted enhancers using reporter gene assays would be of great value for our study. Instead of selecting and testing some enhancer candidates using conventional reporter assays we made use of recently published STARR-seq data sets, which were available via the ENCODE portal (ENCSR064KUD, ENCSR858MPS, (Lee et al., 2020)) for two of our cell lines (K562 and HCT 116). These STARR-seq data sets represent high-throughput plasmid-based reporter assays, which provide an unbiased and quantitative assessment of enhancer activities on a genome-wide scale (Muerdter et al., 2018). We present now as Fig EV3A the proportion of putative enhancers called by TT-seq that overlapped STARR-seq peaks (10% for K562 and 14% for HCT 116). We then compared this to STARR-seq overlaps of putative enhancers called by other methods (DHS, H3K4me1, and H3K27ac) and sets of randomly sampled genomic regions. The proportions of TT-seq enhancers active in STARR-seq were >6-fold enriched over random background, larger than the proportions observed for DHS or H3K4me1 called enhancers, and similar to enhancers defined by H3K27ac. Of note, the proportions of intragenic vs intergenic enhancers overlapping STARR-seq enhancers were lower for all methods (about 1.6-fold on average). We further show that by taking a conservative subset of putative enhancers by combining DHS, H3K27ac and TT-seq, the STARR-seq overlap increased by 4%. Finally, we hypothesized that one of many reasons why some putative enhancers are not active in STARR-seq might be that they are part of cooperating enhancer clusters and may not function when tested individually. To investigate this, we counted the number of neighboring enhancers within ± 12.5 kb for each enhancer that was active in STARR-seq vs not active in STARR-seq. We found that putative enhancers not active in STARR-seq were significantly enriched for enhancers with neighboring enhancers in close proximity (Fig EV3B). We have added a new paragraph to the main text “Comparison to high-throughput functional assays”, in which we describe the STARR-seq comparison (page 10, lines 306-324):

“To further evaluate the functionality of TT-seq defined putative enhancers we examined their activities in genome-wide plasmid-based reporter assays. To this end, we used HCT 116 and K562 STARR-seq data available from the ENCODE portal and investigated the overlap between TT-seq defined enhancers and STARR-seq peaks (Methods). About 14% and 10% of TT-seq defined enhancers were active in STARR-seq for HCT 116 and K562, respectively (Fig EV3A). Next, we compared these results to STARR-seq overlaps of putative enhancers called by other methods and sets of randomly sampled genomic regions (Methods). The proportions of TT-seq enhancers with STARR-seq activity were larger than the proportions observed for open chromatin (DHS) or H3K4me1 defined enhancers, similar to H3K27ac defined enhancers, and more than 6-fold enriched over background (Fig EV3A). We further show that taking a conservative subset of putative enhancers by combining DHS, H3K27ac and TT-seq, increased the STARR-seq overlap to 18% and 14% for HCT 116 and K562, respectively (Fig EV3A). We hypothesized that some enhancers are not active in STARR-seq as they might be part of cooperating enhancer clusters and thus might not function when tested individually. To investigate this, we counted the number of neighboring enhancers within ± 12.5 kbp for each putative enhancer and found that enhancers not active in STARR-seq were significantly enriched for enhancers with neighboring enhancers located in close proximity (Fig EV3B).”

Furthermore, we compared our data to two recently published high-throughput CRISPRi screens in K562 cells, in which they investigated the functionality of ENCODE-predicted enhancers (Gasperini et al., 2019; Schraivogel et al., 2020). In these CRISPRi screens candidate enhancers were epigenetically perturbed and tested in their native genomic

context. They report that only about 4% (Schraivogel et al.) and 10% (Gasperini et al.) of the tested enhancers were functional in their screens. Here, we show that their tested enhancers were significantly more likely to be functional if they overlapped a TT-seq defined putative enhancer region. Moreover, the proportion of CRISPRi validated enhancers also called active by STARR-seq was 25% (Gasperini et al.) and 14% (Schraivogel et al.), respectively. If we take these observed overlaps as reference, it indicates that the observed STARR-seq overlap obtained for TT-seq candidate enhancers is good. We report this finding in the same paragraph “Comparison to high-throughput functional assays” (page 10, lines 325-334):

“To extend these results we asked whether ENCODE-predicted enhancers with validated functionality in K562 cells, as determined by CRISPRi screens (Gasperini et al., 2019; Schraivogel et al., 2020), were enriched for TT-seq defined enhancers. Indeed, enhancers tested in either of the two studies were more likely to be functional if they overlapped a TT-seq defined enhancer (Fisher’s exact test, P -values < 0.01). Finally, we took the verified functional enhancer regions from both studies and calculated the overlap with the K562 STARR-seq peaks. The proportion of CRISPRi verified functional enhancers also called active by STARR-seq was 25% and 14% for Gasperini et al. and Schraivogel et al., respectively. Together these results corroborate the putative enhancers defined by TT-seq in our study.”

Finally, we added a paragraph to the discussion addressing the results obtained from the new main text paragraph “Comparison to high-throughput functional assays” (page 17, lines 561-577):

“To further assess the functionality of the putative enhancers described here, we compared them to available data from two types of large-scale functional assays: genome-wide plasmid-based reporter gene assays (STARR-seq; (Davis et al., 2018)) and high-throughput CRISPRi screens (Gasperini et al., 2019; Schraivogel et al., 2020). In HCT 116 and K562 cells, 14% and 10% of TT-seq defined enhancer regions overlapped a region with STARR-seq activity. Overlaps increased to 18% and 14%, respectively, when we used a more constrained set of candidate enhancers, which were annotated by several enhancer marks (DHS, H3K27ac and TT-seq). Overall this represents a good overlap since even for CRISPRi validated enhancers in K562 (Gasperini et al., 2019; Schraivogel et al., 2020) less than one-fourth (162/677) were active in STARR-seq. Many possible reasons exist for why a CRISPRi-validated enhancer would fail to show activity in a reporter gene assay, in which enhancers are tested outside of their endogenous chromatin environment. One could be that some of the enhancers are part of interacting enhancer clusters and might show no or low reporter gene activity when tested individually. In agreement with this, we found that TT-seq defined putative enhancers not active in STARR-seq were enriched for enhancers with neighboring enhancers located in close proximity (Fig EV3B).”

4. It would be helpful for the reader if the authors explain their rationale for selecting TSS +/- 500 bp for histone modification overlap, to call eRNA. As we know, usually polymerase pauses by 60 bp from the TSS adjacent to the +1 nucleosome and I wonder if the authors have tried various size windows before deciding on 500 bp?

We agree with the reviewer that a more detailed explanation regarding the chosen overlap region will be helpful for the reader. We have not tried various window sizes before making this decision. We added the following sentence to the Methods section (page 24, lines 787-790):

“The ± 500 bp window was chosen to be more permissive, due to the limited resolution of ChIP-seq and since for very lowly expressed transcripts the starts of TT-seq defined TUs may deviate from the actual TSS.”

5. Minor point: It will be also nice to explain the rationale for annotating -500 to +750 of the asRNA as the enhancer region. Also, the space between the divergent TSS is ~300 bp (Tippens et al. Biorxiv 2019, Core et al. Nat. Genet. 2014), so I wonder why did the authors choose a limit of 750 bp as the gap between divergent TSS of intergenic eRNAs.

We agree with the reviewer that this rationale needs a more detailed explanation. ((Core et al., 2014), Fig. 5a, GRO-cap data) and ((Duttke et al., 2015), Fig. 1c, 5'-GRO-seq data) plot the distribution of distances between divergent TSSs. In Core et al. the average distance was 110 bp with max distance around 400 bp, whereas in Duttke et al. the average distance was about 200 with max distance ~700-750 bp. We decided to take a distance cutoff at the permissive end and thus chose 750 bp.

Intragenic enhancers most likely possess also divergent TSSs, but with TT-seq we can judge their transcription only based on the antisense eRNA signal, as the sense TT-seq signal (in the region upstream antisense relative to the annotated antisense RNA) will be a mixture of reads coming from mRNA and eRNA. Since we allowed a gap of max 750 bp for bidirectional intergenic eRNAs, we decided to stick to the 750 bp also as the region upstream of antisense eRNAs, to be consistent. It was again a permissive choice, but in Fig 1 (previously H & I, now E & F (right panels)) it is shown that choosing -750 bp to +500 bp as putative enhancer region actually fits well with the corresponding DNase I-seq and TF ChIP-seq data. Please note that here, in the main text, and in the methods the region is defined as -750 bp to +500 bp relative to the TSS on the antisense strand (i.e. from 750 bp upstream to 500 bp downstream), whereas in the illustration shown in Fig 1C it was previously marked relative to the sense strand as -500 bp to +750 bp. We apologize if this led to confusion and have adapted Fig 1C to make it clearer.

We added now an explanation to the Methods section elucidating the rationale for choosing these cutoffs and boundaries (page 24, lines 801-804; page 25, lines 818-820):

“The permissive 750 bp gap width cutoff was based on 5'-GRO-seq data presented in ((Duttke et al., 2015), Fig. 1c) where they report the distribution of distances between divergent TSSs to be ~200 bp on average and ~700-750 bp maximal. (...) Since we allowed a gap of maximum 750 bp between bidirectional intergenic eRNAs, we selected 750 bp also as putative enhancer region upstream of antisense eRNAs.”

6. In Figure 3B, the authors show variable enhancer transcription across different cell types. However, have they verified if there is any genomic translocation occurring at this locus, resulting in loss of the enhancer region in any of these lines?

We thank the reviewer for pointing this out. We did not verify if there are any genomic translocations occurring at the shown locus, resulting in loss or gain of the enhancer regions in any of the cell lines. In general, we observe a high correlation between the number of putative enhancers annotated in the shown region surrounding MYC and transcription of the MYC gene (Spearman's rho = 0.69). To make the readers aware that we cannot rule out the existence of genomic translocations occurring at the shown locus we added a note of caution to the respective figure legend (Fig EV2):

“Note: TT-seq reads were mapped against the hg38 reference genome and we cannot rule out the existence of genomic rearrangements (e.g. translocations) affecting the shown region in the used cancer cell line clones.”

7. I am confused with Figure 3C. Is this browser shot from WT or deleted cells? Also, for Figure 3D what are the 3 bands for the top panel? At 32 days the gene for the MYC deletion appears to get depleted compared to GAPDH. Please explain.

We thank the reviewer for pointing out that the content of Fig 3C and D needs to be presented more clearly. The genome browser view in Fig 3C (now Fig EV2A) shows the TT-seq signal in wild-type cells. The highlighted region labeled as “CRISPR/Cas9 deletion Myc Δ 2-520/ Δ 2-520” corresponds to the 520 kb region that is deleted in CRISPR/Cas9 deletion cells. The labeling seems to be misleading and we deleted it from the Figure.

CRISPR/Cas9 deletion in the DU 145 cell line was carried out using multiple guides (8 guides) designed across the 520 kb region of the MYC enhancer region (see figure below). The combination of four terminal guides (G1, G2, G3 and G4) can lead to different length of the MYC enhancer deletion region, which can be subsequently detected with genotyping primers (725, 741, 810, 826 bp). The three bands visible on the top panel in Figure 3D correspond to the expected genotyping results. We added a more detailed description of the genotyping protocol to the Methods section (page 21, lines 709-711, 713):

“Note, the combination of the four terminal sgRNAs can lead to different length of the MYC enhancer deletion region, which can be subsequently detected with genotyping primers (725, 741, 810, or 826 bp PCR product). (...) sgRNA locations at the MYC enhancer locus is shown in (Dave et al., 2017).”

We agree with the reviewer that at 32 days the MYC deletion genotyping signal declines slightly. However, we interpret this decline as not significant enough to exclude technical bias due to e.g. pipetting errors or the lower sensitivity of PCR results visualized by gel electrophoresis compared to signals from e.g. digital PCR.

Comparing the genotyping signal in DU 145 deletion cells with the strong decline in signal in GP5d deletion cells (see Figure below from (Dave et al., 2017)) from 16 days onwards, we think it is a valid assumption that the MYC enhancer deletion has no effect on cell proliferation in DU 145 cells. Of course, we cannot completely exclude that the deletion might cause a very small defect in proliferation, which only becomes detectable after culturing the cells for more than 30 days. In this regard we would like to add that we also genotyped the cells at 24 days, which we did not include in the final figure as the signal from the IGH control was missing (see Source Data for Fig EV2). At 24 days the genotyping signal does not indicate a depletion of deletion cells from the cell population.

From (Dave et al., 2017) Figure 6:

(c)

Crispr-Cas9 mediated deletion of region corresponding to Myc Δ 2-540/42-540 in human GP5d colon cancer cells, results in a loss of the edited cells over time. Top panel shows the active enhancer elements in GP5d cells within this region as determined by ChIP-seq analysis of histone H3 lysine 27 acetylation (H3K27ac). The sites of sgRNAs (black lines) and genotyping primers (blue arrows) used are indicated (not to scale). Red arrows mark the enhancer regions

used in this study. Bottom panel shows the PCR-genotyping of the MYC locus and the control IGH locus showing the specific loss of the cells with the edited MYC locus over time. GAPDH was used as internal control. The right panel in each set shows absence of any deletion in the non-transfected cells (day 2). 100 bp ladder DNA molecular weight marker is shown (M).

8. In line 296 the authors claim that TT-seq based enhancer transcription can be a good proxy for enhancer activity. Figure 3 and the text suggest that the enhancer containing genomic region is necessary for the expression of MYC in GP5d cells and its deletion does not cause depletion of DU145 cells where the enhancer is untranscribed. However, this particular region might not be necessary for DU145 cell survival irrespective of its transcriptional status, and MYC expression could be dependent on other enhancers as well. This does not address if enhancer transcription is a good read-out of enhancer activity or not. A more direct way to address this is by doing a CRISPRi of this enhancer region to repress its transcription (for example, in GP5d cells) and then measure the MYC expression/cell proliferation. This will indicate if enhancer transcription itself is necessary for MYC expression and cell proliferation.

We agree with the reviewer that the CRISPR/Cas9 deletion experiment does not address if enhancer transcription is in general a good read-out of enhancer activity and if thus TT-seq is a good proxy of enhancer activity. These conclusions were not our intention and we apologize for the unfortunate wording especially in the concluding sentence. The assumption that enhancer transcription can be a good proxy for enhancer activity is the ‘working model’ of our work and was already shown by others (see introduction lines 109-118 and respective references). The aim of the CRISPR/Cas9 deletion experiment was to show that TT-seq reliably measures enhancer transcription at functionally verified enhancer regions, as opposed to genomic regions with no enhancer function. The genomic region upstream of MYC is characterized by strong TT-seq signal in GP5d cells and was suggested to contain an enhancer region that might be necessary for MYC expression in these cells (Dave et al., 2017). We then show that in DU 145 cells this region shows almost no TT-seq signal and seems to be unnecessary for DU 145 cell survival (CRISPR/Cas9 experiment), and might, therefore, not contain an active enhancer element regulating MYC expression in this cell line. As we do not address if enhancer transcription is in general necessary for enhancer function, we believe the suggested CRISPRi experiment is not needed in this context. We now describe the experiment’s aim and results more clearly (page 9, lines 287-301) and rephrased the paragraph’s concluding sentence to (page 9, lines 301-303):

“In summary, our data captures precise maps of enhancer transcription at well-studied and functionally verified enhancers, and thus TT-seq is a well-suited method to measure enhancer transcription.”
Instead of “(...) and thus enhancer transcription as measured by TT-seq is a good proxy of enhancer activity”.

9. For Figure 4E, is the Med1 ChIP-seq data not available for HCT116 and therefore the authors had to show the H3K27ac as super-enhancer mark? How well could this distinguish super-enhancers from enhancers?

As suspected by the reviewer there was no published Med1 ChIP-seq data available for HCT 116. Thus, we had to use H3K27ac ChIP-seq data instead. According to the first super-enhancer publication (Whyte et al., 2013) Med1 signal is better suited than H3K27ac to distinguish super-enhancers from enhancers, but it is nevertheless widely accepted in the field (and also done by the authors of Whyte et al.) to use H3K27ac when Med1 data is not

available (e.g. (Hnisz et al., 2013)). Since we downloaded the super-enhancer annotation shown in Fig 4E from the SEdb super-enhancer database (Jiang et al., 2019) and didn't do the super-enhancer calling ourselves, we did not check how well H3K27ac separates typical from super-enhancers in this particular data set. We have clarified this now in the respective figure legend (Fig 4E):

"(...) H3K27ac-defined super-enhancer annotation, which was downloaded from SEdb (Jiang et al., 2019). At date of publication no Med1 ChIP-seq data was available for HCT 116."

10. It is interesting to see how cell-specific enhancers drive cell-specific gene expression. With respect to Figure 5D, there have been examples where an enhancer does not activate the nearest promoter rather acts on one at a particular distance. In addition to the current analysis, I wonder if the authors could use the high contact frequency target promoters for the exclusive enhancers (based on Hi-C/capture Hi-C, for available cell lines) instead of the nearest ones and measure target promoter transcription in the those cell lines.

We thank the reviewer for this good suggestion. For LoVo and K562 cell lines we used available promoter capture Hi-C and Pol II ChIA-PET data, respectively, to perform the suggested analysis. In accordance with the results shown in Fig 5D, promoters contacting exclusive enhancers were significantly higher expressed in the respective cell line versus the other cell lines. These results are now presented in Appendix Fig S4D,E and addressed in the main text with the following sentence (page 13, lines 410-413):

"Results were overall similar when instead of analyzing nearest promoters we selected promoters with observed physical proximity to the exclusively transcribed enhancers (Appendix Fig S4D, E)."

11. The authors have provided a nice multi-way strategy to identify E-P pairs, that have been expanded to identify the pairing information of cancer-associated genes. As they have mentioned that many such programs exist to predict target genes for enhancers, it will be interesting to see how their system compares with methods like the ABC model for relevant cell-types (Fulco et al., 2019) that uses enhancer activity (from DHS and H3K27ac peaks) and Hi-C contacts to predict target genes in a cell-type specific manner.

This is an interesting question and we agree with the reviewer that a comparison between our E-P pairing strategies and E-P pairs predicted by the ABC model will be of interest for the reader. To this end we downloaded published ABC model predictions for K562 cells and selected putative enhancers investigated in both data sets. Since one of our pairing approaches paired enhancers to the nearest active gene promoter (within 500 kb), almost all of the enhancers reported in our study paired to at least one promoter. This resulted in a considerable fraction (59%) of enhancers which were paired in our study but not paired by the ABC model. Thus, we decided to focus the comparative analysis on positive E-P pairs reported by the ABC model and compared them to putative E-P pairs described in our study for the same enhancers. For these enhancers, the majority (64%) of the putative E-P pairs reported in our study are supported by the ABC model. We have added the following sentences to the main text (page 14/15, lines 473-480):

"Next, we compared our E-P pairing approach with the activity-by-contact (ABC) model (Fulco et al., 2019), which predicts E-P interactions based on Hi-C contact frequency between enhancer and promoter, and DHS and H3K27ac signal as proxy for enhancer activity. We selected all putative enhancers that were paired to at least one promoter by both our strategy and the ABC model, and compared the overlap between the predicted E-P connections in K562 cells. Using this subset of enhancers, we found that the majority (64%) of the putative E-P pairs reported here were also predicted by the ABC model."

In general, we think the ABC model is a great approach to obtain putative E-P pairs. Since it incorporates Hi-C contact frequencies in the model it provides a strict set of putative pairs with high confidence. However, it is at the same time limiting as Hi-C data might not be sequenced deep enough to support all interactions or data is often not available for a given cell line (and taking Hi-C data from other cell lines also doesn't seem to be an ideal choice). We have added a more general paragraph to the discussion clearly explaining the implications and limitations of the E-P pairing approaches used in our study (page 18, lines 578-602):

“Accurate identification of target genes regulated by enhancers remains challenging. Several computational approaches have been proposed (reviewed in (Hariprakash & Ferrari, 2019)) and their applicability depends on the genomic data sets that are available. Since there are enhancers known to regulate genes over long distances, even spanning regions in the Mb range, and/or to skip over neighboring genes, the target space to look for putative target promoters of an enhancer becomes very large. Pairing enhancers to every gene within a certain distance (e.g. here, ± 500 kbp) would result in a high fraction of false positive predictions (Fulco et al., 2019) and strategies to decrease the number of false positive E-P pairs include considering only pairs with observed physical proximity (Fulco et al., 2019; Javierre et al., 2016) or with correlated activity (Andersson et al., 2014; Thurman et al., 2012). The fraction of physically interacting E-P pairs with correlated transcriptional activity was reported to be about 50% (Fitz et al., 2020). We decided to use a correlation cutoff for all putative E-P pairs that paired beyond the nearest active gene (“correlated window” approach), as physical interaction data was not available for most of the cell lines we investigated. Moreover, the strength of our approach is that TT-seq simultaneously quantifies the transcriptional activity of both gene promoters and enhancers. This allowed us to use correlation of transcriptional activity between enhancers and promoters across all samples to infer tens of thousands of putatively functional E-P pairs in an unbiased way (Fig 6). The pairs we derived are depleted for housekeeping genes and enriched for transcription factors, cancer-associated genes, and 3D conformational proximity (Fig 6). Finally, we provide Table EV5, which contains all possible E-P pairs within ± 500 kbp of each putative enhancer together with the observed distances and correlations of transcriptional activity, providing flexibility to adjust distance and/or correlation constraints.”

12. Overall, TT-seq based enhancer annotation and characterization is a significant addition to other genome-wide methods for mapping enhancers, especially in so many different cancers. However, it will be interesting to see a comparative analysis or overlap between enhancers called by TT-seq vs GRO-cap/ChRO-seq or CAGE in relevant cell lines.

We thank the reviewer for the good suggestion. We obtained putative enhancer annotations called using ChRO-seq data in Jurkat cells and called using CAGE data for four of the cell lines (K562, Jurkat, DU145, and PC3). As expected, we observed much higher sensitivity in calling transcribed enhancers from TT-seq (and ChRO-seq) data compared to CAGE data. Of the putative enhancers annotated by CAGE approximately half were also annotated by TT-seq (or ChRO-seq). Furthermore, about half of the putative Jurkat enhancers reported here overlapped with putative enhancers called by ChRO-seq. We present this overlap analysis now in Appendix Fig S1E and reference it in the text as follows (page 6/7, lines 206-205):

“The resulting putative enhancer regions (hereafter also referred to as ‘enhancer regions’ or ‘enhancers’) were highly concordant with DNase I hypersensitive sites (Fig 1E; (Dunham et al., 2012)) and transcription factor binding regions identified by ChIP-seq (Fig 1F; (Yan et al., 2013)), and overlapped with enhancer annotations derived from CAGE or ChRO-seq data (Appendix Fig S1F; (Andersson et al., 2014; Chu et al., 2018)). As expected, TT-seq was more sensitive in calling transcribed enhancers than CAGE.”

Reviewer #2:

This manuscript reports a set of TT-seq data obtained from a collection of cancer cell lines. The emphasis is on genome-wide identification of putative enhancers from these data. The data appear to be of high quality, and the analyses are generally carefully done and clearly described. I found the manuscript easy to read and the figures are very clear. The logic that is followed is not very much out-of-the-box, and the insights obtained are a bit incremental. Nevertheless, there are some interesting observations and the data will be a very useful resource for the community.

We thank the reviewer for seeing the contribution of our work and for the constructive suggestions below.

Major points:

1. The elephant in the room is: does TT-seq really do a good job identifying functional enhancers? This is not a new problem: one can ask the same for most enhancer predictions based on histone marks, ATAC-seq, Hi-C methods, and MPRAs. Very few of these data have been properly benchmarked against a real gold standard, which is to delete a representative set of the predicted enhancers by CRISPR and measure the activity of the surrounding promoters. The current manuscript does not go further than anecdotal (and perhaps somewhat cherry-picked) support provided on pages 8-9.

I think it would go too far to demand a systematic CRISPR deletion test for the current manuscript, because it could easily take a year of work to do this for a sufficient number of predicted enhancers. But it would be good if the authors could emphasize this limitation, and resist the temptation to over-interpret their results, for example in title, abstract, bottom page 6, and bottom page 9, line 476, and the Discussion. It may be good to systematically add the word "putative" or "candidate" in front of the word "enhancer".

We agree with the reviewer that this limitation needs to be clearly emphasized. We have now very carefully addressed these limitations in the text and rephrased respective parts. We have made the following changes:

- *abstract: we systematically added the word "putative" in the abstract to emphasize that all reported enhancers and E-P pairs are putative.*
- *text: We wish to not systematically add the word "putative" or "candidate" in front of every word "enhancer". Instead, we have clearly highlighted in the second paragraph that the presented putative eRNA/enhancer regions are thereafter referred to as eRNA/enhancer regions (page 6, lines 183 and 199), and for all key concluding sentences at the ends of each paragraph we have now systematically added the word "putative" or "candidate" in front of the words "enhancer" or "E-P pair" (below we only list the sentences pointed out by the reviewer):*
 - *page 6 (now page 7, lines 204/205): "These efforts identified thousands of **putative eRNAs and enhancers** in fourteen human cancer cell lines (Table EV1)."*
 - *page 9: we deleted the respective sentence "TT-seq, therefore, reliably distinguished active from inactive enhancer regions."*
 - *line 476 (now page 16, lines 517-520): "To aid studies of cancer genomics, we created a comprehensive catalog of transcribed **putative** enhancers with cancer-associated somatic mutations (Table EV3) and **putative** enhancer-promoter pairs involving cancer genes from the Cancer Gene Census (Sondka et al., 2018) (Table EV4)."*
- *discussion: we systematically added the word "putative"*

In addition, we think that our detailed comparisons to high-throughput functional data (STARR-seq reporter assays and high-throughput CRISPR screens; see below) can serve as a partial benchmark to help the reader to get a clearer picture of what to expect from our candidate enhancer annotations.

Perhaps as a partial benchmarking, the authors could overlay their K562 data with the CRISPR screen that Schraivogel et al, Nat Meth 2020 recently reported? They found only 4% of ENCODE-predicted "enhancers" to control a nearby gene, and it will be interesting to know whether TT-seq has added value in predicting functional enhancers.

We thank the reviewer for this good suggestion. We compared our data to two recently published high-throughput CRISPRi screens in K562 cells, in which they investigated the functionality of ENCODE-predicted enhancers, the one suggested by the reviewer (Schraivogel et al., 2020) and (Gasperini et al., 2019). In these CRISPRi screens candidate enhancers were epigenetically perturbed and tested in their native genomic context. They report that only about 4% (Schraivogel et al.) and 10% (Gasperini et al.) of the tested enhancers were functional in their screens. Here, we show that their tested enhancers were significantly more likely to be functional if they overlapped a TT-seq defined putative enhancer region, indicating added value of TT-seq in predicting functional enhancers.

We also compared the TT-seq candidate enhancers, as well as the functionally verified enhancers from before (Gasperini et al. & Schraivogel et al.), to available STARR-seq data for two of the cell lines (please refer to our detailed response to reviewer #1, comment 3). Considering that the proportion of CRISPRi validated enhancers also called active by STARR-seq was 25% (Gasperini et al.) and 14% (Schraivogel et al.), we find a good overlap between TT-seq candidate enhancers and regions active in STARR-seq (Fig EV3).

We have added a new paragraph to the main text "Comparison to high-throughput functional assays", in which we describe the comparisons to STARR-seq and the CRISPRi screens (page 10):

"To further evaluate the functionality of TT-seq defined putative enhancers we examined their activities in genome-wide plasmid-based reporter assays. To this end, we used HCT 116 and K562 STARR-seq data available from the ENCODE portal and investigated the overlap between TT-seq defined enhancers and STARR-seq peaks (Methods). About 14% and 10% of TT-seq defined enhancers were active in STARR-seq for HCT 116 and K562, respectively (Fig EV3A). Next, we compared these results to STARR-seq overlaps of putative enhancers called by other methods and sets of randomly sampled genomic regions (Methods). The proportions of TT-seq enhancers with STARR-seq activity were larger than the proportions observed for open chromatin (DHS) or H3K4me1 defined enhancers, similar to H3K27ac defined enhancers, and more than 6-fold enriched over background (Fig EV3A). We further show that taking a conservative subset of putative enhancers by combining DHS, H3K27ac and TT-seq, increased the STARR-seq overlap to 18% and 14% for HCT 116 and K562, respectively (Fig EV3A). We hypothesized that some enhancers are not active in STARR-seq as they might be part of cooperating enhancer clusters and thus might not function when tested individually. To investigate this, we counted the number of neighboring enhancers within ± 12.5 kbp for each putative enhancer and found that enhancers not active in STARR-seq were significantly enriched for enhancers with neighboring enhancers located in close proximity (Fig EV3B).

To extend these results we asked whether ENCODE-predicted enhancers with validated functionality in K562 cells, as determined by CRISPRi screens (Gasperini et al., 2019; Schraivogel et al., 2020), were enriched for TT-seq defined enhancers. Indeed, enhancers tested in either of the two studies were more likely to be functional if they overlapped a TT-seq defined enhancer (Fisher's exact test, P-values < 0.01). Finally, we took the verified functional enhancer regions from both studies and calculated the overlap with the K562 STARR-seq peaks. The proportion of CRISPRi verified functional enhancers

also called active by STARR-seq was 25% and 14% for Gasperini et al. and Schraivogel et al., respectively. Together these results corroborate the putative enhancers defined by TT-seq in our study.”

Finally, we added a paragraph to the discussion to discuss the results obtained from the new main text paragraph “Comparison to high-throughput functional assays” (page 17, lines 561-577):

“To further assess the functionality of the putative enhancers described here, we compared them to available data from two types of large-scale functional assays: genome-wide plasmid-based reporter gene assays (STARR-seq; (Davis et al., 2018)) and high-throughput CRISPRi screens (Gasperini et al., 2019; Schraivogel et al., 2020). In HCT 116 and K562 cells, 14% and 10% of TT-seq defined enhancer regions overlapped a region with STARR-seq activity. Overlaps increased to 18% and 14%, respectively, when we used a more constrained set of candidate enhancers, which were annotated by several enhancer marks (DHS, H3K27ac and TT-seq). Overall this represents a good overlap since even for CRISPRi validated enhancers in K562 (Gasperini et al., 2019; Schraivogel et al., 2020) less than one-fourth (162/677) were active in STARR-seq. Many possible reasons exist for why a CRISPRi-validated enhancer would fail to show activity in a reporter gene assay, in which enhancers are tested outside of their endogenous chromatin environment. One could be that some of the enhancers are part of interacting enhancer clusters and might show no or low reporter gene activity when tested individually. In agreement with this, we found that TT-seq defined putative enhancers not active in STARR-seq were enriched for enhancers with neighboring enhancers located in close proximity (Fig EV3B).”

2. I wonder what the impact is of the decision of the authors to discard all TT-seq signals that overlap with protein-coding genes. I understand that this may be done because of overlapping gene transcription, but surely there are a lot of enhancers in introns that are missed this way? A rough estimate of the "blind" proportion of the genome would be helpful.

This seems to be a misunderstanding and we apologize for not stating this clearly enough in the manuscript. With TT-seq we annotate intergenic AND intragenic enhancers/eRNAs. The only difference is that intergenic enhancers are defined based on bidirectional non-coding transcription from both strands, whereas intragenic enhancers are defined based on the signal from transcription antisense of the protein-coding gene only (asRNAs; excluding regions within ± 1 kb of protein-coding TSSs; see Fig 1B,C and Appendix Fig S1B), since TT-seq signal from the sense strand will be a mixture of transcription from the protein-coding gene and transcription of the intragenic enhancer. The proportion of intragenic and intergenic enhancers we observe is 50/50 on average, which is in line with previous reports. We have rephrased the respective results section of the manuscript to make this clearer and also state the percentage of TT-seq defined intragenic and intergenic enhancers (page 6, lines 191-198):

“To define intergenic enhancer regions, we selected the region between the TSSs of each pair of intergenic bidirectional eRNAs and extended it by 500 bp in both directions (Fig 1C, Methods). To define intragenic enhancer regions, we selected the region covering 750 bp upstream to 500 bp downstream of the TSS of antisense eRNAs. This annotation strategy led to the identification of on average ~8,000 (3,848 – 12,337) putative enhancer regions per cell line (Fig 1D), of which about half were located within protein-coding genes (Appendix Fig S1B).”

Furthermore, we show now in Appendix Fig S1B-E that important marks to annotate enhancers are enriched around the TSS of antisense eRNAs relative to the gene-body of the mRNAs, allowing us to confidently annotate them as intragenic eRNAs.

3. Likewise, the E-P pairing by correlation of the TT-seq signals lacks "hard" experimental

validation. It is quite possible that many of these correlations are only correlations and not causal E->P effects. Again, this would require a lot of CRISPR deletions of predicted enhancers to check the impact on the predicted partner promoter. If it is the choice of the authors not to do such validations, then of course the value of this analysis remains to be seen. This caveat should then be emphasized clearly in the text, because certainly not everyone in the cancer field may be aware of it.

We thank the reviewer for this critical comment. We agree that E-P pairing by correlation lacks "hard" experimental validation, as already mentioned earlier by the reviewer the main problem is a missing gold standard of true E-P pairs to be used for validation. As a rough estimate a recent study (Fitz et al., 2020) showed, albeit based on the assumption that interacting (based on Pol II ChIA-PET) E-P pairs are truly functional E-P pairs, that about 50% of E-P pairs had correlated enhancer and promoter transcription (based on GRO-seq signal). Since for most of the cell lines investigated here there is currently no Hi-C (or other) interaction data available, we decided to use a correlation cutoff for all E-P pairs not involving the nearest active gene. This allowed us to include putative pairings beyond the nearest active gene and at the same time to reduce false positive predictions by not allowing all possible interactions. When we investigated how many putative E-P pairs within ± 500 kb were supported by ChI-C in LoVo cells if no correlation cutoff was used, the fraction dropped from 27% (Fig 6F) to 16%. Thus, we think imposing a correlation cutoff on non-nearest E-P pairs is a reasonable choice. However, this means that we may miss some true interactions (not involving the nearest active gene and being less or not correlated) and may include some pairs which are correlated but do not represent causal E->P effects. Overall, we agree with the reviewer that not everyone reading this manuscript may be aware of the limitations imposed by the used pairing strategies and thus we added the following paragraph to the discussion (page 18, lines 578-602):

"Accurate identification of target genes regulated by enhancers remains challenging. Several computational approaches have been proposed (reviewed in (Hariprakash & Ferrari, 2019)) and their applicability depends on the genomic data sets that are available. Since there are enhancers known to regulate genes over long distances, even spanning regions in the Mb range, and/or to skip over neighboring genes, the target space to look for putative target promoters of an enhancer becomes very large. Pairing enhancers to every gene within a certain distance (e.g. here, ± 500 kbp) would result in a high fraction of false positive predictions (Fulco et al., 2019) and strategies to decrease the number of false positive E-P pairs include considering only pairs with observed physical proximity (Fulco et al., 2019; Javierre et al., 2016) or with correlated activity (Andersson et al., 2014; Thurman et al., 2012). The fraction of physically interacting E-P pairs with correlated transcriptional activity was reported to be about 50% (Fitz et al., 2020). We decided to use a correlation cutoff for all putative E-P pairs that paired beyond the nearest active gene ("correlated window" approach), as physical interaction data was not available for most of the cell lines we investigated. Moreover, the strength of our approach is that TT-seq simultaneously quantifies the transcriptional activity of both gene promoters and enhancers. This allowed us to use correlation of transcriptional activity between enhancers and promoters across all samples to infer tens of thousands of putatively functional E-P pairs in an unbiased way (Fig 6). The pairs we derived are depleted for housekeeping genes and enriched for transcription factors, cancer-associated genes, and 3D conformational proximity (Fig 6). Finally, we provide Table EV5, which contains all possible E-P pairs within ± 500 kbp of each putative enhancer together with the observed distances and correlations of transcriptional activity, providing flexibility to adjust distance and/or correlation constraints."

4. There seems to be some circular reasoning in the manuscript:

a. l181-182: "The region surrounding the TSS of eRNAs was enriched for the active enhancer mark H3K27ac" -- yes, but l169-174 describe that eRNAs were required to

originate from sites defined by a set of histone marks, which surely are not independent from H3K27ac.

This is a valid comment and if we would have called enhancer states based on histone marks in each of the cell lines that were used here, we would fully agree with the reviewer. But since we use a universal set of enhancer states predicted from histone marks (not including H3K27ac) in 127 cell types, which did not include the cell lines used here (except for K562), we think that enrichment for H3K27ac (and TFs) is not a given fact and thus worth to report (especially for a cell line that is not K562, namely LoVo).

b. l183: "and for transcription factor binding as monitored by ChIP-seq" -- same issue, ChIP-seq peaks of TFs are strongly correlated with histone marks.

Please, see response regarding a).

c. eRNAs were required to be bi-directional (l75-l176). Surely this causes a bias in the directionality scores (l239-...)?

We agree with the reviewer. Though it is clearly stated in the text that the observed directionality scores are for bidirectional enhancers. We have added a note to the respective legend of Fig 2D addressing this (page 47, lines 1846/1847):

"Note that the distribution would be less balanced if unidirectionally transcribed enhancers were included."

Bas van Steensel - review assignment 27 July 2020; completed 10 Aug 2020, apologies for the delay! [It is my standard policy to sign and date ALL of my reviews, regardless of my opinions and recommendations. PLEASE DO NOT REMOVE THIS NOTE.]

Reviewer #3:

The paper by Lidschreiber et. al. analyzed TT-seq data in 14 cancer cell lines. The data all appears to be excellent quality, and was sequenced to a high read depth. Most of the data provided is new data. The data analysis was conducted in an expert manner, providing an atlas of transcription units, enhancers, and enhancer targets that will be used elsewhere. As a result, this manuscript describes an excellent resource that is very clearly written and will be an important contribution to the literature. I am in favor of publication, however I have a few comments that should be addressed:

We thank the reviewer very much for seeing the value and contribution of our work, and for the constructive suggestions below.

* All of the cell lines studied were model cell lines, and some have been passaged for a very long time in tissue culture. The authors state that transcriptional signatures are in good agreement with primary tumors. While this is no doubt true to some extent, it's clear that there are a lot of differences between primary tumors and cell lines as well due to differences in microenvironment and genetic factors. This does not diminish my enthusiasm for the present study, but I would like the authors to discuss the disadvantages (and advantages!) of these cell models in more detail.

We agree with the reviewer that this is an important point to discuss. We have now addressed this point by extending the "cancer" paragraph in the discussion section (page 18/19, lines 603-629):

“Our comprehensive annotation of putative enhancers and E-P pairs in human cancer cell lines provides a valuable tool for further studying enhancer function in cancer. *Tumor-derived cell lines carry many of the original oncogenic alterations* and transcriptional profiles of these cell lines have been shown to be in good agreement with those of primary tumors (Ghandi et al., 2019; Greshock et al., 2007; Iorio et al., 2016). However, it is important to consider that cancer cell line models have major limitations and inevitably entail genomic differences as pointed out by several studies (Ertel et al., 2006; Gillet et al., 2011; Sandberg & Ernberg, 2005; Stein et al., 2004). Cell culturing conditions fail to recapitulate the condition of tumor cell growth in patients and cell line models cannot accurately model the complexity of the patient tumor microenvironment, i.e. interaction with stroma and immune cells. Nevertheless, cancer cell lines are an indispensable tool in cancer research and preclinical studies of anticancer drugs (Goodspeed et al., 2016; Katt et al., 2016; Mirabelli et al., 2019; Sharma et al., 2010). Cancer cell lines can be readily manipulated, allow global and detailed mechanistic studies, and thereby provide insights into the molecular mechanisms of tumor growth. Recently, several studies have characterized the genome wide molecular profiles of cancer cell lines and investigated how well they model the genomic profiles, molecular subtypes, and heterogeneity of tumors (Barretina et al., 2012; Domcke et al., 2013; Garnett et al., 2012; Ghandi et al., 2019; Holliday & Speirs, 2011; Klijn et al., 2015). Together these results allow researchers to select cancer cell lines that best model the genomic features of a particular type of tumor. The large-scale whole-genome sequencing projects of cancer cell lines and cancer patient samples such as the Cancer Cell Line Encyclopedia (Ghandi et al., 2019) and the Pan-Cancer Analysis of Whole Genomes (Campbell et al., 2020) have constructed large catalogs of somatic mutations in many types of tumors. Integrating such data and extracting correlations between enhancer activity and tumor growth will be a valuable task for the future.

* The manuscript identified only ~3000-3500 enhancers in each cell line. This seems somewhat low based on reports from other assays. Is this a matter of sensitivity? Can the authors describe factors that could influence sensitivity? Related to this, TT-seq does not get any signal from a paused RNA polymerase. Could this decrease sensitivity?

This seems to be a misunderstanding and we apologize for not stating this clearly enough in the manuscript. In Fig 1 and the respective text we state the number of eRNAs per cell line (previously Fig 1D, now Figure EV1A). In the revised manuscript, we have also added a plot showing the number of enhancers per cell line (new Fig 1D), ~8000 on average. These numbers are lower than putative enhancers defined by open chromatin or histone marks, higher than transcribed putative enhancers reported by CAGE, and are in line with what has been reported by other assays that define putative enhancers based on enhancer transcription. E.g. two publications describing GRO/PRO-seq defined enhancers report enhancer numbers which are higher (Danko et al., 2015) and lower (Franco et al., 2018) in numbers compared to ours, respectively. Of course, one has to keep in mind that the number of enhancers varies strongly between cell types and that the number of transcription-defined enhancers is strongly dependent on the expression cutoff, sequencing depth, and annotation pipeline used, making detailed comparisons difficult. Nevertheless, it is a valid question to ask if the fact that TT-seq does not get signal from paused Pol II (as compared to GRO/PRO/ChRO-seq) decreases its sensitivity to detect transcribed enhancers. In line with what reviewer #1 requested (12.) we compared TT-seq defined enhancers with enhancers defined by ChRO-seq in Jurkat cells (Appendix Fig S1F). The number of enhancers called by ChRO-seq was about 2-fold higher, suggesting that a fraction of enhancers might be engaged with paused Pol II but without detectable transcriptional activity. It has to be noted though, that the dREG pipeline used to call regulatory regions from this ChRO-seq data also calls unidirectionally transcribed intergenic regulatory regions, thus making a direct comparison difficult.

* The authors use a cutoff directionality score >3 as imbalance and preferentially transcribed

on both strands. However, this directionality score appears to be defined in log-10 units. It seems as though a lot of the features with a score <3 may also be quite imbalanced, depending on one's perspective. Could the authors please clarify, or update their description.

We thank the reviewer for pointing this out and thereby avoiding misunderstandings. This was wrongly stated in the legend of Fig 2A,D. The directionality score was not defined in log-10 units, but log10 transformed for plotting in Fig 2A,D. We set the cutoff for the directionality score to 3-fold difference between strands. We have now made it clear in the respective figure legend that the directionality score cutoff of 3 is defined in linear space and that the histogram shows log10 transformed directionality scores. We agree that depending on one's perspective a difference by e.g. 2.5-fold can seem already imbalanced. The reason we chose 3-fold as cutoff is simply that we wanted to do it in accordance to a recent publication by Struhl and Churchman labs (Jin et al., 2017). In their study, they showed that promoter regions are intrinsically bidirectional and are shaped by evolution to bias transcription toward coding versus non-coding RNAs. We added a sentence to make clearer that we chose 3-fold as cutoff to match what was used in Jin et al. (page 47, lines 1841-1848):

"Histogram shows genome-wide distribution of log10-transformed directionality scores at bidirectional enhancers. Directionality score for enhancers is defined as the ratio of reads mapped to the plus and minus strand. Only bidirectional intergenic enhancer regions shorter than 3 kbp were considered (Methods). In accordance with (Jin et al., 2017) enhancer regions were categorized as having a preferred direction (white) if the absolute value of the plus to minus strand ratio was ≥ 3 and as bidirectionally balanced (grey) if it was <3."

- Core, L. J., Martins, A. L., Danko, C. G., Waters, C. T., Siepel, A., & Lis, J. T. (2014). Analysis of nascent RNA identifies a unified architecture of initiation regions at mammalian promoters and enhancers. *Nature Genetics*, 46(12), 1311-1320. <https://doi.org/10.1038/ng.3142>
- Danko, C. G., Hyland, S. L., Core, L. J., Martins, A. L., Waters, C. T., Lee, H. W., . . . Siepel, A. (2015). Identification of active transcriptional regulatory elements from GRO-seq data. *Nature Methods*, 12(5), 433-438. <https://doi.org/10.1038/nmeth.3329>
- Dave, K., Sur, I., Yan, J., Zhang, J., Kaasinen, E., Zhong, F., . . . Taipale, J. (2017). Mice deficient of Myc super-enhancer region reveal differential control mechanism between normal and pathological growth. *eLife*, 6, 1-25. <https://doi.org/10.7554/eLife.23382>
- Dunham, I., Kundaje, A., Aldred, S. F., Collins, P. J., Davis, C. A., Doyle, F., . . . Lochovsky, L. (2012). An integrated encyclopedia of DNA elements in the human genome. *Nature*, 489(7414), 57-74. <https://doi.org/10.1038/nature11247>
- Duttke, S. H. C., Lacadie, S. A., Ibrahim, M. M., Glass, C. K., Corcoran, D. L., Benner, C., . . . Ohler, U. (2015, Feb 19). Human promoters are intrinsically directional. *Mol Cell*, 57(4), 674-684. <https://doi.org/10.1016/j.molcel.2014.12.029>
- Fitz, J., Neumann, T., Steininger, M., Wiedemann, E. M., Garcia, A. C., Athanasiadis, A., . . . Pavri, R. (2020, May). Spt5-mediated enhancer transcription directly couples enhancer activation with physical promoter interaction. *Nat Genet*, 52(5), 505-515. <https://doi.org/10.1038/s41588-020-0605-6>
- Franco, H. L., Nagari, A., Malladi, V. S., Li, W., Xi, Y., Richardson, D., . . . Kraus, W. L. (2018, Feb). Enhancer transcription reveals subtype-specific gene expression programs controlling breast cancer pathogenesis. *Genome Res*, 28(2), 159-170. <https://doi.org/10.1101/gr.226019.117>

- Fulco, C. P., Nasser, J., Jones, T. R., Munson, G., Bergman, D. T., Subramanian, V., . . . Engreitz, J. M. (2019, Dec). Activity-by-contact model of enhancer-promoter regulation from thousands of CRISPR perturbations. *Nat Genet*, *51*(12), 1664-1669. <https://doi.org/10.1038/s41588-019-0538-0>
- Gasperini, M., Hill, A. J., McFaline-Figueroa, J. L., Martin, B., Kim, S., Zhang, M. D., . . . Shendure, J. (2019). A Genome-wide Framework for Mapping Gene Regulation via Cellular Genetic Screens. *Cell*, *0*(0), 1-14. <https://doi.org/10.1016/J.CELL.2018.11.029>
- Hnisz, D., Abraham, B. J., Lee, T. I., Lau, A., Saint-André, V., Sigova, A. A., . . . Young, R. A. (2013). Super-enhancers in the control of cell identity and disease. *Cell*, *155*(4), 934-947. <https://doi.org/10.1016/j.cell.2013.09.053>
- Jiang, Y., Qian, F., Bai, X., Liu, Y., Wang, Q., Ai, B., . . . Li, C. (2019). SEdb: a comprehensive human super-enhancer database. *Nucleic Acids Research*, *47*(D1), D235-D243. <https://doi.org/10.1093/nar/gky1025>
- Jin, Y., Eser, U., Struhl, K., & Churchman, L. S. (2017, Aug 24). The Ground State and Evolution of Promoter Region Directionality. *Cell*, *170*(5), 889-898 e810. <https://doi.org/10.1016/j.cell.2017.07.006>
- Kundaje, A., Meuleman, W., Ernst, J., Bilenky, M., Yen, A., Heravi-Moussavi, A., . . . Principal, i. (2015, 2015/02/01). Integrative analysis of 111 reference human epigenomes. *Nature*, *518*(7539), 317-330. <https://doi.org/10.1038/nature14248>
- Lee, D., Shi, M., Moran, J., Wall, M., Zhang, J., Liu, J., . . . Gerstein, M. (2020). STARRPeaker: Uniform processing and accurate identification of STARR-seq active regions. *bioRxiv*, 694869. <https://doi.org/10.1101/694869>
- Muerdter, F., Boryn, L. M., Woodfin, A. R., Neumayr, C., Rath, M., Zabidi, M. A., . . . Stark, A. (2018, Feb). Resolving systematic errors in widely used enhancer activity assays in human cells. *Nat Methods*, *15*(2), 141-149. <https://doi.org/10.1038/nmeth.4534>
- Schraivogel, D., Gschwind, A. R., Milbank, J. H., Leonce, D. R., Jakob, P., Mathur, L., . . . Steinmetz, L. M. (2020, Jun). Targeted Perturb-seq enables genome-scale genetic screens in single cells. *Nat Methods*, *17*(6), 629-635. <https://doi.org/10.1038/s41592-020-0837-5>
- Whyte, W. A., Orlando, D. A., Hnisz, D., Abraham, B. J., Lin, C. Y., Kagey, M. H., . . . Young, R. A. (2013). Master transcription factors and mediator establish super-enhancers at key cell identity genes. *Cell*, *153*(2), 307-319. <https://doi.org/10.1016/j.cell.2013.03.035>

Thank you for sending us your revised manuscript. We have now heard back from the three reviewers who were asked to evaluate your study. As you will see the reviewers are satisfied with the modifications made and think that the study is now suitable for publication.

Before we can formally accept your manuscript, we would ask you to address a few remaining editorial issues listed below.

REFEREE REPORTS

Reviewer #1:

The authors have thoroughly addressed all the points raised in the original manuscript. In particular, I am glad to see the comparison of the putative enhancers to the functional assays and also the comparative analyses they have done to other prediction methods, that highlight the advantages of using transcription data while high quality Hi-C data is not available for all cell lines. Regarding the overlap of the ChRO-seq vs TT-seq annotated enhancers, it is interesting to see that ChRO-seq is calling many more putative enhancers than TT-seq, but the authors have clearly mentioned the limitations of such comparisons due to inherent differences in the calling approaches that is helpful for the readers. Overall this work is of high-quality and I highly recommend publishing it.

Reviewer #2:

All my concerns/suggestions have been addressed. I recommend publication.

Bas van Steensel - review assignment accepted 17 Nov 2020; completed 20 Nov 2020 [It is my standard policy to sign and date ALL of my reviews, regardless of my opinions and recommendations. PLEASE DO NOT REMOVE THIS NOTE.]

Reviewer #3:

My comments have been adequately addressed. Many thanks to the authors for their contributions to the scientific literature!

The authors have made all requested editorial changes.

Thank you again for sending us your revised manuscript. We are now satisfied with the modifications made and I am pleased to inform you that your paper has been accepted for publication.

Corresponding Author Name: Michael Lidschreiber, Patrick Cramer

Manuscript Number: MSB-20-9873